



# Extension of a gaseous dry deposition algorithm to oxidized volatile organic compounds and hydrogen cyanide for application in chemistry transport models

Zhiyong Wu[1,2], Leiming Zhang[1,*], John T. Walker[3], Paul A. Makar[1], Judith A. Perlinger[4], Xuemei Wang[5]

[1]Air Quality Research Division, Science and Technology Branch, Environment and Climate Change Canada, Toronto, ON, M3H 5T4, Canada

[2]ORISE Fellow at US Environmental Protection Agency, National Risk Management Research Laboratory, Research Triangle Park, NC, 27711, USA

[3]US Environmental Protection Agency, National Risk Management Research Laboratory, Research Triangle Park, NC, 27711, USA

[4]Civil & Environmental Engineering Department, Michigan Technological University, Houghton, MI, 49931, USA

[5]Institute for Environmental and Climate Research, Jinan University, Guangzhou, 510632, China

*Correspondence to: Leiming Zhang (leiming.zhang@canada.ca)





**Abstract**: With increasing complexity of air quality models, additional chemical species have been
included in model simulations for which dry deposition processes need to be parameterized. For
this purpose, the gaseous dry deposition scheme of Zhang et al. (2003) is extended to include 12
oxidized volatile organic compounds (oVOCs) and hydrogen cyanide (HCN) based on their
physicochemical properties, namely the effective Henry's law constants and oxidizing capacities.
Modeled dry deposition velocity ($V_d$) values are compared against field flux measurements over a
mixed forest in the southeastern U.S. during June 2013. The model captures the basic features of
the diel cycles of the observed $V_d$. Modeled $V_d$ values are comparable to the measurements for
most of the oVOCs at night. However, modeled $V_d$ values are mostly around 1 cm s$^{-1}$ during
daytime, which is much smaller than the observed daytime maxima of 2-5 cm s$^{-1}$. Analysis of the
individual resistance terms/uptake pathways suggests that flux divergence due to fast atmospheric
chemical reactions near the canopy was likely the main cause of the large model-measurement
discrepancies during daytime. The extended dry deposition scheme likely provides conservative
$V_d$ values for many oVOCs. While higher $V_d$ values and bi-directional fluxes can be simulated by
coupling key atmospheric chemical processes into the dry deposition scheme, we suggest that more
experimental evidence of high oVOC $V_d$ values at additional sites is required to confirm the
broader applicability of the high values studied here. The underlying processes leading to high
measured oVOC $V_d$ values require further investigation.



## 1. Introduction

Atmospheric pollutants impact human health and can also cause detrimental effects on sensitive ecosystems (Wright et al., 2018). Quantifying atmospheric deposition for atmospheric pollutants is needed to estimate their lifetimes in air and deposition rates to ecosystems. The amount of dry deposition of a pollutant of interest is typically calculated as the product of its ambient concentration and its dry deposition velocity ($V_d$), with $V_d$ being calculated using empirically developed dry deposition schemes (Wesely & Hicks, 2000). Existing dry deposition schemes are known to have large uncertainties even for the most commonly studied chemical species such as $O_3$, $SO_2$ and more commonly measured nitrogen species with relatively rich flux datasets (Flechard et al., 2011; Wu et al., 2012; Wu et al., 2018).

Existing dry deposition schemes have thus far considered a small number of oxidized volatile organic compounds (oVOCs). Due to the lack of field flux data of oVOCs, $V_d$ of these species is typically parameterized based on physicochemical properties, taking $SO_2$ and $O_3$ as references (Wesely, 1989; Zhang et al., 2003). However, Karl et al. (2010) found that $V_d$ of oVOCs calculated using existing schemes are about a factor of 2 lower than those based on canopy-level concentration gradient measurements over six forest and shrubland sites. $V_d$ in their study was calculated from an inverse Lagrangian transport model with concentration gradient data as model input. The ratio of magnitudes between $V_d$(oVOCs) and $V_d$($O_3$) in the study of Karl et al. (2010) are similar to those of Zhang et al. (2003) in that $V_d$(oVOCs) is slightly smaller than $V_d$($O_3$) in both cases. However, the typical daytime $V_d$($O_3$) over vegetated canopies is around 1 cm s$^{-1}$ in the literature from numerous studies (see summary in Silva & Heald, 2018), and the value in Karl et al. (2010) is much higher (e.g., up to 2.4 cm s$^{-1}$ at canopy top). One hypothesis explaining both high $V_d$($O_3$) and high $V_d$(oVOCs) would be the reaction of $O_3$ with oVOC, which depends on the



chemical structure of the oVOC, but data required for validating this hypothesis are still lacking.
We thus suspect that the very high $V_d$(oVOCs) presented in Karl et al. (2010) were likely caused
by atmospheric chemical processes not typically considered in the dry deposition process. High
$V_d$(oVOCs) values were also observed over a temperate mixed forest in the southeastern U.S. in a
more recent short-term study (Nguyen et al., 2015), which again were suspected to be caused by
atmospheric chemical reactions near vegetation surface. The flux measurements themselves also
contain uncertainty. For example, Wu et al. (2015) showed that different measurement methods
(e.g., flux-gradient versus eddy correlation) resulted in very different daytime $V_d$($O_3$) over the same
forest canopy. Hydrogen cyanide (HCN) is one of the most abundant cyanides present in the
atmosphere (Singh et al., 2003) and is considered a biomass burning marker (Bunkan et al., 2013),
but few existing studies have considered its dry deposition, which is critical to estimating the total
sinks and atmospheric lifetimes of cyanides.

To fulfill community demands of modeling dry deposition of organic compounds (Kelly et

al., 2019; Moussa et al., 2016; Paulot et al., 2018; Pye et al., 2015; Xie et al., 2013) and to take
advantage of the recent flux dataset of a large number of oVOCs and HCN (Nguyen et al., 2015),
the present study extends the current Zhang et al. (2003) scheme by including 12 additional oVOC
species and HCN. The parameterization for these newly-included species is based on the effective
Henry's law constants and oxidizing capacities of the individual species and by considering the
measured $V_d$ values as well. Model-measurement comparison is conducted for $V_d$ as well as
resistance components/uptake pathways, results from which identify the major causes of model-
measurement discrepancies. Model parameters were chosen to produce the magnitude of nighttime
$V_d$ for nearly all the chemical species, but this approach inevitably underpredicted daytime $V_d$
values for several oVOCs species with very high measured daytime $V_d$ values. This approach is

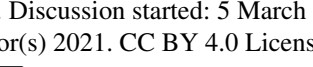



recommended due to the following considerations: (1) some of the chemical processes causing
flux loss at the surfaces may be treated separately in the mass continuity equation in chemical
transport models, (2) some of the oVOCs may also experience bi-directional air-surfaceexchange,
and (3) more flux measurements are needed to confirm if the very high daytime flux for certain
oVOCs is an universal phenomenon, noting that the existing data used here were from a short
period of several days and over only one surface type.

At this stage with very limited knowledge on oVOC $V_d$, air-surface exchange models based

on various theoretical and/or measurement approaches should be developed, so that these models
can be made available to the scientific community where such models are urgently needed, and for
future evaluation and improvement should more flux measurements become available. For
example, Nguyen et al. (2015) modified the Wesely (1989) scheme to fit the flux data, while in
the present study a more theoretically constrained approach was used. A more sophisticated model
for handling air-canopy exchange of semivolatile organic compounds is also available in the
literature (Nizzetto and Perlinger, 2012). Note that a bottom-up approach was adopted in Nizzetto
and Perlinger (2012) to estimate fluxes as compared to the present study, which provides a top-
down determination of deposition velocity through comparison with measured (bottom-up) fluxes.
oVOC $V_d$ values from all the existing models may all be within the uncertainty range.

**2. Methodology**
Dry deposition of a gaseous compound to most canopy types is mainly through nonstomatal uptake
during nighttime and through both nonstomatal and stomatal uptake during daytime. The
nonstomatal uptake depends on water solubility and reactivity of the species, which can be





quantified by its effective Henry's Law constant ($H^*$) and oxidizing capacity, respectively (Wesely,
1989; Zhang et al., 2002). In the Supporting Information (SI) document, Table S1 lists $H^*$ values
and Table S2 lists the oxidizing capacities for oVOCs and HCN considered in the present study.
Following the approach described in Zhang et al. (2002), two model parameters ($\alpha$ and $\beta$) are
needed for every chemical species to calculate the nonstomatal uptake, with $\alpha$ being dependent on
$H^*$ and $\beta$ dependent on oxidizing capacity. Initial $\alpha$ values were first given based the relative
magnitudes of $H^*$ of all the chemical species and that of $SO_2$. Considering that the majority of the
chemical species are very reactive, a value of 1.0 was used for $\beta$ for most species and smaller
values for a few less reactive species. $\alpha$ and $\beta$ values were then adjusted based on the agreement
of nighttime $V_d$ between modeled values and measured fluxes obtained from a forest site in the
southeastern US during summer (Nguyen et al., 2015). When adjusting $\alpha$ and $\beta$ values, two rules
were first applied: (1) the trends in $\alpha$ (or $\beta$) values between different chemical species should be
consistent with the trends of their log($H^*$) (or oxidizing capacity) (see Figure S1 for the finalized
$\alpha$ versus log($H^*$)); and (2) modeled mean and median nighttime $V_d$ should be mostly within a
factor of 2.0 of the measured values (see discussion in Section 3.2 below). Only after these two
rules were satisfied, then the possible maximum $\alpha$ and $\beta$ values were chosen to reduce the gap
between the modeled and measured daytime $V_d$, knowing that model predicted $V_d$ were mostly
lower than the measured ones. Model theory and field data used for model evaluation are briefly
described below.

*2.1. Parameterization scheme for $V_d$ of oVOCs and HCN*
The gaseous dry deposition scheme of Zhang et al. (2003) (hereinafter referred to as the Model)
was originally designed to model $V_d$ for 31 chemical compounds including 9 inorganic species and





22 organics. Formic acid (HCOOH) is the only oVOC species that is available in both the Model
and the flux measurement dataset used here (described in the next section). In this study, the Model
was extended to include 12 new oVOC species and HCN. Briefly, $V_d$ is calculated according to:
$$V_d(z) = \left( R_a(z) + R_b + R_c \right)^{-1},\tag{1}$$

where $R_a$ is the aerodynamic resistance, $R_b$ the quasi-laminar sub-layer resistance, $R_c$ the surface
resistance, and $z$ the reference height above the vegetation. $R_a$ and $R_b$ can be estimated using the
conventional micrometeorological approaches based on similarity theory and the equations used
in the Model can be found in Wu et al. (2018). $R_c$ is parameterized as:
$$\frac{1}{R_c} = \frac{1 - W_{st}}{R_s + R_m} + \frac{1}{R_{ns}},\tag{2}$$

$$\frac{1}{R_{ns}} = \frac{1}{R_{ac} + R_g} + \frac{1}{R_{cut}},\tag{3}$$

where $R_s$ is the canopy stomatal resistance, $R_m$ the mesophyll resistance, $R_{ns}$ the non-stomatal
resistance including resistance for uptake by leaf cuticles ($R_{cut}$) and by soil or ground litter ($R_g$),
$R_{ac}$ in-canopy aerodynamic resistance, and $W_{st}$ the fraction of stomatal blocking under wet
conditions.

$R_s$ for any gaseous oVOCs ($i$) or HCN is also calculated using the sunlit/shade stomatal

resistance approach as was done in Zhang et al. (2002):
$$\frac{1}{R_{s,i}} = G_s(PAR) f(T) f(D) f(\Psi) \frac{D_i}{D_{H_2O}}.\tag{4}$$

Here $G_s(PAR)$ is the unstressed canopy stomatal conductance for water vapor, a function of
photosynthetically active radiation ($PAR$). The dimensionless functions $f(T)$, $f(D)$ and $f(\psi)$ range
from 0 to 1, representing the fractional degree of stomatal closure caused by the stress from





temperature, water vapor pressure deficit, and leaf water potential, respectively. $D_{H_2O}$ and $D_i$ are
the molecular diffusivities for water vapor and the gas of interest, respectively.

$R_{cut}$ and $R_g$ for any oVOCs or HCN are scaled to those of $SO_2$ and $O_3$ with two species ($i$)-

dependent scaling parameters $\alpha(i)$ and $\beta(i)$:

$$\frac{1}{R_{cut/g}(i)} = \frac{\alpha(i)}{R_{cut/g}(SO_2)} + \frac{\beta(i)}{R_{cut/g}(O_3)} \ . \tag{5}$$

$\alpha(i)$ and $\beta(i)$ values for all the chemical species are listed in Table 1, which are assigned using the
method described above.

The $R_m$ for HCN was set to 100 s m$^{-1}$ based on its effective Henry's law constants and

oxidizing capacities. Karl et al. (2010) found that enzymatic conversion can be an efficient pathway
for the immobilization of oVOCs (e.g., methacrolein and methyl vinyl ketone, acetaldehyde,
methacrolein) within leaf interior, besides dissolution and oxidation, which suggests that the
magnitude of $R_m$ for oVOCs is minimal. Thus, the $R_m$ for the oVOCs was set to 0 s m$^{-1}$ (Table 1).

*2.2. Field flux data*
The fluxes of 16 atmospheric compounds (including 13 oVOC species, HCN, hydrogen peroxide
($H_2O_2$), and nitric acid ($HNO_3$)) were measured using the eddy covariance (EC) technique at the
Centreville ("CTR") Southeastern Aerosol Research and Characterization Study (SEARCH) site
(hereinafter referred to as CTR). The CTR site (Brent, Alabama; 32.90°N, 87.25°W) is surrounded
by a grassy field to the south and a temperate mixed forest that is part of the Talladega National
Forest in all the other directions. The forest canopy is comprised of needleleaf coniferous (shortleaf,
longleaf, and loblolly pine; ~40%) and broadleaf deciduous (primarily oak, sweetgum, and hickory;
~60%) tree species. A 20 m metal walk-up tower is used as the main structure supporting



instruments that measured the eddy covariance fluxes and related meteorological variables. The
sonic anemometer and the gas inlet were mounted at a height of about 22 m, facing north toward
the forest. The canopy height near the tower is on average ~10 m with a leaf area index (LAI) of
~4.7 $m^2$ $m^{-2}$. A database of half-hourly $V_d$ for 16 atmospheric compounds covering 5 non-
continuous days in June 2013 was obtained at the site. During these periods, the predominant winds
were northerly which is ideal to sample air from the forest (Figure S2) and the requirement on
energy balance closure was met (see Nguyen et al. (2015)). At CTR, it was typically humid (RH
50-80%) and warm (28-30 °C) in the daytime during the experiment (Figure S3). A comprehensive
description of the $V_d$ dataset, data processing protocols, the instrumental methods, uncertainty
analysis, and the site characterizations can be found in Nguyen et al. (2015).

**3. Results and Discussion**
*3.1. Comparison of modeled resistance components*
*3.1.1. Atmospheric resistances ($R_a$ and $R_b$)*
For very reactive and soluble substances such as $HNO_3$ and $H_2O_2$, $R_c$ is often assumed to be close
to 0 (Hall & Claiborn, 1997; Meyers et al., 1989; Valverde-Canossa et al., 2006; Wesely & Hicks,
2000). The analysis of the measurement data showed that the daytime averaged $V_d$ for $HNO_3$ and
$H_2O_2$ fitted well the rate of deposition without surface resistance ($V_d = 1/[R_a+R_b]$) (Nguyen et al.,
2015), which supports the assumption of near zero $R_c$ for $HNO_3$ and $H_2O_2$ over the mixed
deciduous-coniferous CTR site under humid environment. Therefore, the measured $V_d$ of $HNO_3$
and $H_2O_2$ can be used to evaluate the modeled atmospheric resistances for those species (the sum
of $R_a$ and $R_b$). $R_a$ represents the resistance for turbulent transport between the reference height and





the surface and is not chemical compound specific. $R_b$ quantifies the resistance for the mass transfer
across the thin layer of air in contact with surface elements and is a function of the molecular
diffusivity of a specific compound (Wesely & Hicks, 1977). In theory, the differences in $R_b$
between any two gaseous species are only determined by differences in their molecular diffusivity
at any given turbulent condition.

Figure 1 compares the modeled average diel variations of $V_d$ for $HNO_3$ and $H_2O_2$ against

observations. The measured $V_d$ for $HNO_3$ and $H_2O_2$ peaked around noon at about 4 cm s$^{-1}$ and 6
cm s$^{-1}$, respectively, and were less than 1 cm s$^{-1}$during the night. The model reproduced the diel
pattern and captured the peak $V_d$ values at noon well. During the early night time (hours 19-23),
the modeled $V_d$ for $HNO_3$ and $H_2O_2$ were on the order of 1 cm s$^{-1}$, much higher than the
measurements (<0.2 cm s$^{-1}$). During the night, $R_a$ dominates atmospheric resistance as it is usually
much larger than $R_b$ in magnitude. This discrepancy between the measurement and the model
during the early night could be due to the stability correction functions used in the $R_a$ calculation
(the equations can be found in the article by Wu et al. (2018)) which is subject to large uncertainties
under nocturnal stable conditions (Högström, 1988). The measurements indicated that $H_2O_2$
deposited slightly faster than $HNO_3$, and the model reproduces well, as shown in Figure 1. Modeled
$R_b$ for $H_2O_2$ is always smaller than that for $HNO_3$ due to the smaller molecular weight and the larger
molecular diffusivity. Overall, the model was in good agreement with the measurements regarding
$V_d$ for $HNO_3$ and $H_2O_2$, implying that the parameterization for atmospheric resistances ($R_a$ and $R_b$)
was reasonable for the site during the study period.

*3.1.2. Stomatal resistance ($R_s$)*



Over vegetative areas, gas molecules can exit and enter the leaf through the stomata by molecular
diffusion, similar to the leaf-air exchange of water vapor and $CO_2$. In dry deposition models, $R_s$
for water vapor is estimated using evapotranspiration stomatal submodels, an approach that is also
popular in the land surface and climate communities. $R_s$ is extended to any gas species using the
ratio of molecular diffusivity of the species of interest to that of water vapor (Pleim & Ran, 2011;
Wesely & Hicks, 2000). Figure 2 compares the modeled canopy stomatal conductance ($G_s = 1/R_s$)
for water vapor against the observation-based estimates. The observation-based $G_s$ was estimated
by using the inversion of the Penman-Monteith (P-M) equation (Monteith & Unsworth, 1990)
which calculates $R_s$ for water vapor by using measured water vapor fluxes and related
meteorological data (e.g., humidity, temperature). The evaporation from soil water and liquid
water on the vegetation surfaces is usually a minor contribution to the total water vapor flux
observed above a forest canopy during summer daytime. It was assumed that 85% of the water
vapor flux originated from transpiration in this study, following that used in the study of
Turnipseed et al. (2006) at Duke Forest, North Carolina. Note that a value of 90% was used by
Clifton et al. (2017) at Harvard Forest, Massachusetts. The uncertainty of the calculated $R_s$ related
with the uncertainty in water vapor flux portion (on an order of 10%) is much smaller than the
differences between the modeled and the observation-based stomatal conductance (by a factor of
two) as discussed below.

The model reproduced the basic diel pattern in $G_s$ (i.e., highest values between 08:00 and

11:00) but the peak value is only about half of the observation-based values. The Jarvis stomatal
submodel (Jarvis, 1976) used in the Model is known for its linear dependence on the prescribed
minimum stomatal resistance ($R_{s,min}$), a term that is subject to large uncertainties (Kumar et al.,
2011; Wu et al., 2018; Wu et al., 2011). A series of  tests conducted by iteratively adjusting the





$R_{s,min}$ values showed the modeled $G_s$ to be in better agreement with observations if $R_{s,min}$ was
decreased by 40% (Figure 2). $G_s$ from the Model with the adjusted $R_{s,min}$ was in good agreement
with the observation-based values for most of the time, though the modeled values were slightly
smaller than the observation-based estimates around noon. Analysis of the $R_s$ parameterization
indicates that this discrepancy was related to the stress function for water vapor pressure deficit
(VPD) used in the Jarvis stomatal submodel, which may overpredict the stress on stomatal opening
due to high VPD around noon.

*3.1.3. Non-stomatal resistance ($R_{ns}$)*
To assess if the non-stomatal resistance ($R_{ns}$) parametrization (Eq. 3) is reasonable, modeled $1/R_{ns}$
(defined as $G_{ns}$) values are compared with the non-stomatal portion of the flux, the inverse of
which is termed the residual conductance ($G_{residual}$). $G_{residual}$ includes all processes influencing
deposition aside from $R_a$, $R_b$, $R_m$, and $R_s$, calculated as $[V_d^{-1} - (R_a + R_b)]^{-1} - (R_s + R_m)^{-1}$. Here $V_d$ is
from the observations, $R_a$ and $R_b$ are calculated using the Model driven by the observed
meteorology, $R_s$ is the observation-based estimates by the P-M method, adjusted by the molecular
diffusivity of each gas (similar to Eq. 4), and $R_m$ is listed in Table 1. Although considerable
uncertainties in the calculated $G_{residual}$ exist (in this form of back-calculation, we must assume that
the $G_{ns}$ terms are correctly estimated), it can provide useful information on the flux/$V_d$ resulting
from processes such as deposition to the leaf cuticle and ground (i.e., non-stomatal) or chemical
loss due to reactions within and near the canopy that lead to flux divergence.

Figure 3 compares the observation-based $G_{residual}$ for each oVOC species or HCN against

the corresponding modeled non-stomatal conductance ($G_{ns}$) under different conditions. The mean
and median values are presented in Table S3. During the nighttime when the canopy surface was



dry (no dew), the $G_{residual}$ for oVOC species ranged from 0.08 to 0.18 cm s$^{-1}$ and the modeled $G_{ns}$
was comparable in magnitude. When the surface was wet from dew formation on leaves and
needles, the oVOC species showed an increase in $G_{residual}$ by 55%-440% compared to the nighttime
dry surface. The model captured the increases in non-stomatal uptake when the surface become
wet with dew, although it may underestimate (e.g., HDC$_4$, INP, HCN) or overestimate (e.g., PAA,
DHC$_4$, HCOOH) the wetness effects. During the daytime of the study period, no precipitation was
recorded at the CTR site (Figure S3) and the canopy surface was dry. The mean $G_{residual}$ for oVOCs
ranged from 0.5 cm s$^{-1}$ to 8.7 cm s$^{-1}$ during the daytime, much higher than the modeled $G_{ns}$ for most
species (0.2 - 1 cm s$^{-1}$). Figure S4 presents the diel variations of $G_{residual}$ and $G_{ns}$ and it shows that
the modeled $G_{ns}$ showed smaller diel variations than those of $G_{residual}$ and large differences in
magnitude can be seen during the daytime. The modeled $G_{ns}$ showed a peak during the early
morning (around 7:00) which may be due to the enhanced non-stomatal uptake by dew wetted
surfaces.


*3.2. Evaluation of modeled deposition velocities*
Figure 4 shows model-measurement comparison of diel $V_d$ of the oVOCs and HCN and Table 2
presents the statistical results of the comparison. As described in Section 2, the assigned $\alpha$ and $\beta$
values should first produce reasonable nighttime $V_d$. Modeled nighttime mean $V_d$ were very close
to measurements for the majority of the chemical species, although the differences were somewhat
larger for the median values (Table 2). Three species (HAC, HPALD, PROPNN) still had 50%
lower modeled than measured nighttime mean $V_d$, but have slightly higher modeled than measured
nighttime median $V_d$. In contrast, modeled daytime mean $V_d$ were more than 50% lower than the



measured values for four species (HMHP, PAA, HPALD, ISOPOOH/IEPOX) and were also
significantly lower for several other species. Only three species (MTNP, HCN, HCOOH) had
comparable modeled and measured $V_d$ for both day- and nighttime. One species (DHC$_4$) had
slightly lower of modeled than measured daytime mean or median $V_d$, but with an opposite trend
for nighttime $V_d$.

The model reproduced the basic features of the diurnal pattern of the observations, showing

highest values during the day and lowest values at night. Correlation coefficients between the
measurement and the model ranged from 0.52 to 0.77. At night, the measured $V_d$ for the oVOCs
remained relatively low, typically ranging from 0.1-0.5 cm s$^{-1}$, and the model produced the same
magnitudes for most of the species. During the daytime, the model can only capture the magnitudes
of the measured $V_d$ for a few species (e.g., HCN, HCOOH, MTNP, DHC$_4$), of which the peak $V_d$
values were less than 1.5 cm s$^{-1}$. For the other species, the measured peak $V_d$ values were in the
range of 2 to 5 cm s$^{-1}$, while the modeled results were below 1 cm s$^{-1}$. As shown in section 3.1.2,
the modeled $G_s$ was likely underestimated when compared to the simultaneous measurements of
water vapor flux. Adjusting $G_s$ higher by 67% (through reducing $R_{s,min}$ by 40%) can only increase
the modeled $V_d$ of the oVOCs by 10-40% during the daytime (see the sensitivity test in Figure 4),
and the peak values were still mostly below 1 cm s$^{-1}$. Figure 5 shows that the model captured the
differences in measured $V_d$ for the oVOCs to some extent. The model-measurement agreements
were good for species with the measured mean $V_d$ below 0.5 cm s$^{-1}$, above which the discrepancy
increased. For the measurements, the mean values were significantly larger than the median values,
especially for the fast-deposited species, indicating that the distribution of the measured $V_d$ values
skewed to the right (high values). The model has a better agreement with the measurements by
comparing the median versus mean values.



At night when stomata are mostly closed and atmospheric chemical reactions are largely
inhibited, the measured fluxes above the canopy should better represent non-stomatal surface
uptake. In the presence of sunlight, fast chemical reactions between the inlet and canopy could
make a significant or even dominant contribution to the measured fluxes of reactive species
(Farmer & Cohen, 2008; Wolfe et al., 2011). The impact of fast chemical reactions on surface
fluxes should be different for different chemical species. To verify this hypothesis, two chemical
species (HAC and PAA) having similar molecular weights (74 Da and 76 Da, respectively) but
very different daytime fluxes were compared (Figure 6). Their similar molecular diffusivities
(controlled by molecular weight) suggest that they should be transferred through the quasi-laminar
sub-layer and taken up through leaf stomata at similar rates, resulting in similar resistance
components of $R_b$ and $R_s$. Note that $R_a$ is universal to any trace gases and $R_m$ is assumed to be
negligible. Thus, the differences between their $V_d$ should be caused by their different non-stomatal
sinks. At night, $V_d$ values were similar between HAC and PAA (median values: 0.04 cm s$^{-1}$) over
dry surfaces. When the surfaces were wet due to dew formation, $V_d$ for both HAC and PAA
increased (median values: 0.30-0.48 cm s$^{-1}$). In contrast, $V_d$(PAA) was much higher than $V_d$(HAC)
during daytime, suggesting additional or larger sinks exist for PAA compared to HAC. Thus, fast
chemical processing and subsequent flux divergence above the canopy likely caused the large
discrepancies between the measured and modeled $V_d$ for the reactive oVOC compounds during the
daytime.
Chemical processes indeed can cause flux divergence or convergence at the surface, which
has been supported by growing evidence from field measurements (e.g., Farmer and Cohen, 2008;
Min et al., 2014; Wolfe et al. 2009). For example, Wolfe et al. (2009) suggested that the differences
in loss rate between the inlet and canopy may be an important contributor to the measured net flux





of peroxyacetyl nitrate, irrespective of turbulent timescales. Photochemical OH production is
reduced within canopies, which in turn slows down the oxidation of volatile organic compounds
and the photolysis of organic nitrates. The oVOCs measured at the CTR site are mainly produced
from the oxidation of isoprene and monoterpenes (Nguyen et al., 2015). Most of the oVOCs are
quite chemically reactive and can undergo fast oxidation (e.g., multifunctional carbonyls),
decomposition (e.g., HMHP), or photolysis (e.g., organic nitrates) (Müller et al., 2014; Nguyen et
al., 2015). Vertical gradients in the chemical production and loss rates below the inlet can exhibit
chemical flux divergence, which contributes to the net flux above canopy. Quantifying the effects
of chemical processing on the net flux would require a multi-layer model with resolved emission,
deposition, turbulent diffusion, and chemical processes throughout the canopy, which is
recommended for future studies  (e.g., Ashworth et al., 2015; Bryan et al., 2012; Stroud et al., 2005;
Wolfe & Thornton, 2011; Zhou et al., 2017).

Quantifying $V_d$ as the ratio of flux to concentration at one measurement height only ($V_d =$

$F/C_{zr}$), rather than as the ratio of flux to the concentration difference at the measurement height
and the surface ($V_d = F/[C_{zr} - C_0]$), although commonly employed in analyzing eddy covariance
flux measurements, is a simplification. It is valid for 1) matter that disappears nearly completely
by reactions at the surface, and 2) unstable or neutral conditions. Most chemical species considered
here may satisfy the first condition. With regards to the second condition, our analysis is based on
the assumption that, under stable conditions at nighttime, concentrations observed at the
measurement height change in relation to the fluxes measured at this height. However, no relation
between measured concentration and flux is typically observed due to the presence of a shallow
stable boundary layer, connection between the stable free atmosphere and stable boundary layer
by internal gravity waves, ground inversions, and low-level jets, leading to intermittent turbulence





at the measurement height containing a gravity wave signal, and non-steady-state conditions
(Foken, 2017). Future efforts to model oVOC and HCN deposition velocities above forest canopies
should be based on neutral or unstable boundary layer flux measurements only, or, for example,
on modified Bowen ratio flux measurement in which concentrations are measured at two heights
in the constant flux layer. Such an approach can provide a means to compute a measured deposition
velocity of a surface reactive substance as proportional to the ratio between the measured flux and
the measured concentration difference.

**4. Summary and recommendations**
The number of chemical species simulated in chemical transport models (CTMs) has been
increasing with increasing computer power. Among these, oVOCs and HCN are an important
groups of atmospheric pollutants for which dry deposition processes need to be treated as
accurately as possible, so that their inputs to ecosystems (noting that some oVOCs are organic
nitrogen) and their roles on other atmospheric chemistry processes (e.g., formation of ozone and
secondary organic aerosols) can be assessed. Earlier dry deposition schemes have considered very
few oVOCs and need to be extended for more species. Dry deposition of HCN was assumed to be
negligible in some CTMs (e.g., Moussa et al., 2016). The present study first generated effective
Henry's law constant and oxidizing capacity, the two key physical and chemical properties that
are considered to control the dry deposition process (Wesely & Hicks, 2000), for 12 oVOCs
species and HCN. Two scaling factors for the non-stomatal resistance and one for the mesophyll
resistance were applied to individual oVOCs and HCN for calculating their respective $V_d$.
The modeled nighttime $V_d$ agrees well with the measured data for most of the oVOCs,
suggesting that the current non-stomatal parameterization scheme is a reasonable approach. The



stomatal conductance for water vapor, with adjusted (reduced) $R_{s,min}$, also agrees well with
measured values. However, the modeled peak $V_d$ values during daytime are only a fraction (0.2-
0.5) of the measured values for some of the oVOCs, suggesting that fast atmospheric chemical
processes likely contributed to the total measured fluxes. In practice, these additional fluxes during
daytime can be modeled as non-stomatal uptake and better model-measurement agreement can be
obtained by adjusting the non-stomatal parameterization scheme (e.g., Müller et al., 2018; Paulot
et al., 2018). However, using this approach will produce unreasonably high values for the solubility
parameter and overpredict $V_d$ during nighttime if the same non-stomatal formulas are used for both
day and nighttime (as is the case in the existing schemes). More importantly, the high measured
$V_d$ have only been observed at relatively few sites during very short periods (Karl et al., 2010;
Nguyen et al., 2015).  More evidence is needed to parameterize $V_d$ for oVOCs to different land use
categories over entire seasons. Until then, the conservative estimates of $V_d$ such as modeled in this
study are still recommended for use in chemical transport models. The model parameters chosen
for $V_d$ of these oVOCs provide the best-known representation of their respective physicochemical
properties, and the modelled $V_d$ values fall within the range of the low-end values of the available
measurements.
Future field studies should focus on conducting flux measurements of oVOC compounds
with highest uncertainties, such as those that are most chemically reactive in the atmosphere or
most rapidly taken up by wet surfaces. Additional measurements are also needed in different
ecosystems to inform the representativeness of the high oVOC $V_d$ reported by Nguyen et al. (2015)
and Karl et al. (2010). Furthermore, concurrent chemical measurements of oxidants such as $O_3$ and
radicals are needed to quantify flux divergence due to fast within-and near-canopy chemical
reactions. Future dry deposition schemes should include additional biochemical processes and





species-dependent parameters for non-stomatal uptake, including enzymatic reactions (Karl et al., 2010), the octanol-air partitioning coefficients to account for the cavity formation and polar intermolecular interactions with leaf surfaces and reservoirs (Nizzetto and Perlinger, 2012), and the enhancement/reduction effects due to soil and leaf moisture. Chemical processes within the canopy airspacecould also be coupled with emission and deposition schemes to realistically simulate chemicals fate and transport, including bi-directional fluxes of reactive compounds discussed here, as well as less reactive compounds such as methanol. Such an approach would require specification of chemical conditions within and near the canopy as well as in-canopy radiation and air flow. While more computationally intensive, the results presented here reinforce the need for such advanced models to explicitly resolve the non-stomatal processes contributing to the net atmosphere-biosphere exchange of reactive compounds. Above all, intercomparison studies should be first conducted for existing models that can handle oVOC dry deposition processes to quantify the magnitudes of uncertainties in the simulated $V_d$ as well as the associated ambient concentration and deposition fluxes.

**Code and data availability**

The computer code and data used in this study can be obtained from containg the corresponding author.

**Competing interests**

The authors declare that they have no conflict of interest.

**Author contributions**

ZW conducted model run and data analysis and drafted the manuscript. LZ designed the project, finalized computer code, drafted part of the manuscript and finalized the paper. JTW contributed



to manuscript writing and commented on the manuscript. PAM generated chemistry data that are
used in the supporting document and commented on the manuscript. JAP contributed to model
design and manuscript writing and commented on the manuscript. XW contributed to the project
design and commented on the manuscript.

**Acknowledgments**
We thank Tran Nguyen for the field flux data and Glenn Wolfe and Christopher Groff for the tree
survey data. We also greatly appreciate helpful comments from Tran Nguyen, Chris Geron and
Donna Schwede. X. Wang was supported by the Chinese National Key Research and Development
Plan (2017YFC0210100) and the State Key Program of National Natural Science Foundation of
China (91644215). The SouthEastern Aerosol Research and CHaracterization (SEARCH) network
was sponsored by the Southern Company and the Electric Power Research Institute. The field data
during    the    SOAS    2013    campaign    is    available    at
https://esrl.noaa.gov/csd/groups/csd7/measurements/2013senex/Ground/DataDownload/.
*Disclaimer: The research presented was not performed or funded by U.S. Environmental*
*Protection Agency and was not subject to EPA's quality system requirements. The views expressed*
*in this article are those of the authors and do not necessarily represent the views or policies of the*
*U.S. Environmental Protection Agency.*

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





Table 1. List of model parameters needed in the scheme of Zhang et al. (2003) for simulating dry deposition velocity of additional oVOCs species and HCN: $\alpha$ and $\beta$ are scaling parameters for non-stomatal resistance, and $R_m$ is mesophyll resistance.

| Symbol | Name | Molecular Weight (Da) | Scaling Parameters | | $R_m$ (s m$^{-1}$) |
|---|---|---|---|---|---|
| | | | $\alpha$ | $\beta$ | |
| HMHP | hydroxymethyl hydroperoxide | 64 | 5 | 1 | 0 |
| HAC | hydroxyacetone | 74 | 1.5 | 1 | 0 |
| PAA | peroxyacetic acid | 76 | 2 | 1 | 0 |
| HDC$_4$ | the C4 hydroxy dicarbonyl from IEPOX oxidation | 102 | 1 | 0.2 | 0 |
| DHC$_4$ | the C4 dihydroxy carbonyl from IEPOX oxidation | 104 | 2 | 0.2 | 0 |
| HPALD | isoprene hydroperoxy aldehydes | 116 | 1.5 | 1 | 0 |
| ISOPOOH/IEPOX [a] | isoprene hydroxyhydroperoxide and isoprene dihydroxyepoxide | 118 | 5 | 0.2 | 0 |
| PROPNN | propanone nitrate or propanal nitrate | 119 | 1.5 | 1 | 0 |
| ISOPN | isoprene hydroxy nitrates | 147 | 1.5 | 1 | 0 |
| MACN/MVKN [a] | methacrolein and Methyl vinyl ketone hydroxy nitrate | 149 | 1.5 | 1 | 0 |
| INP | isoprene nitrooxy hydroperoxide | 163 | 1.5 | 1 | 0 |
| MTNP | monoterpene nitrooxy hydroperoxide | 231 | 1.5 | 1 | 0 |
| HCN | hydrogen cyanide | 27 | 0 | 0.1 | 100 |
| HCOOH [b] | formic acid | 46 | 2 | 0.2 | 0 |

[a] Treated as one group of compounds in the field measurements due to instrument limitation and have the same parameter values in the model.

[b] Beta value for HCOOH in Zhang et al. (2003) is 0.0, and here is given as 0.2 to be consistent to other oVOC species here (which would make no difference since the alpha value of 2 would dominate the nonstomatal resistance).





Table 2. Statistical results of the observed and modeled dry deposition velocity ($V_d$) for oVOCs and HCN (cm s$^{-1}$) [a]

| Compound | All | | | | | Daytime | | | | Nighttime | | |
|---|---|---|---|---|---|---|---|---|---|---|---|---|
| | N | Obs | Mod | Mod-$R_{s,min}$ | R | N | Obs | Mod | Mod-$R_{s,min}$ | N | Obs | Mod |
| HMHP | 247 | 1.66 (0.61) | 0.69 (0.54) | 0.75 (0.58) | 0.63 | 85 | 3.42 (3.49) | 1.05 (1.04) | 1.19 (1.17) | 128 | 0.33 (0.13) | 0.37 (0.24) |
| HAC | 245 | 0.84 (0.53) | 0.41 (0.31) | 0.49 (0.36) | 0.61 | 84 | 1.21 (1.07) | 0.65 (0.62) | 0.81 (0.78) | 128 | 0.44 (0.12) | 0.21 (0.15) |
| PAA | 243 | 1.08 (0.52) | 0.46 (0.34) | 0.53 (0.37) | 0.74 | 85 | 2.18 (2.15) | 0.71 (0.69) | 0.86 (0.83) | 128 | 0.28 (0.09) | 0.24 (0.17) |
| HDC$_4$ | 205 | 0.45 (0.22) | 0.30 (0.20) | 0.37 (0.23) | 0.64 | 66 | 0.91 (0.78) | 0.51 (0.49) | 0.66 (0.65) | 111 | 0.10 (0.06) | 0.15 (0.10) |
| DHC$_4$ | 247 | 0.42 (0.21) | 0.41 (0.31) | 0.47 (0.36) | 0.61 | 85 | 0.92 (0.85) | 0.63 (0.61) | 0.76 (0.73) | 128 | 0.08 (0.06) | 0.22 (0.16) |
| HPALD | 247 | 1.11 (0.46) | 0.39 (0.29) | 0.45 (0.34) | 0.67 | 85 | 2.08 (2.17) | 0.60 (0.58) | 0.73 (0.70) | 128 | 0.40 (0.10) | 0.21 (0.15) |
| ISOPOOH/IEPOX | 247 | 1.02 (0.49) | 0.63 (0.48) | 0.67 (0.52) | 0.59 | 85 | 2.11 (2.06) | 0.94 (0.94) | 1.05 (1.05) | 128 | 0.28 (0.09) | 0.34 (0.23) |
| PROPNN | 246 | 0.89 (0.43) | 0.39 (0.29) | 0.45 (0.33) | 0.53 | 84 | 1.40 (1.38) | 0.60 (0.58) | 0.73 (0.70) | 128 | 0.46 (0.13) | 0.21 (0.15) |
| ISOPN | 247 | 0.68 (0.39) | 0.38 (0.28) | 0.43 (0.33) | 0.62 | 85 | 1.27 (1.29) | 0.58 (0.57) | 0.70 (0.67) | 128 | 0.21 (0.09) | 0.21 (0.15) |
| MACN/MVKN | 246 | 0.65 (0.32) | 0.38 (0.28) | 0.43 (0.32) | 0.57 | 84 | 1.19 (1.15) | 0.58 (0.57) | 0.70 (0.66) | 128 | 0.22 (0.06) | 0.21 (0.15) |
| INP | 247 | 0.64 (0.46) | 0.38 (0.28) | 0.43 (0.33) | 0.63 | 85 | 1.12 (1.17) | 0.57 (0.56) | 0.68 (0.65) | 128 | 0.24 (0.10) | 0.20 (0.15) |
| MTNP | 246 | 0.33 (0.13) | 0.36 (0.27) | 0.40 (0.31) | 0.54 | 84 | 0.55 (0.57) | 0.54 (0.54) | 0.64 (0.62) | 128 | 0.16 (0.04) | 0.20 (0.15) |
| HCN | 234 | 0.13 (0.06) | 0.17 (0.15) | 0.22 (0.20) | 0.77 | 84 | 0.26 (0.24) | 0.33 (0.34) | 0.43 (0.45) | 117 | 0.03 (0.01) | 0.03 (0.01) |
| HCOOH | 244 | 0.47 (0.27) | 0.46 (0.35) | 0.54 (0.41) | 0.52 | 83 | 0.82 (0.75) | 0.72 (0.68) | 0.91 (0.88) | 127 | 0.20 (0.05) | 0.23 (0.16) |

[a] Note: N is the number of samples; R is the correlation coefficient between observation (Obs) and model simulation (Mod); "Mod-$R_{s,min}$" refers to a sensitivity test in which $R_{s,min}$ was reduced by 40%; Daytime is 09:00-17:00 (local time) and nighttime is 20:00-06:00 (local time). Median values are provided in parentheses, following arithmetic mean values.





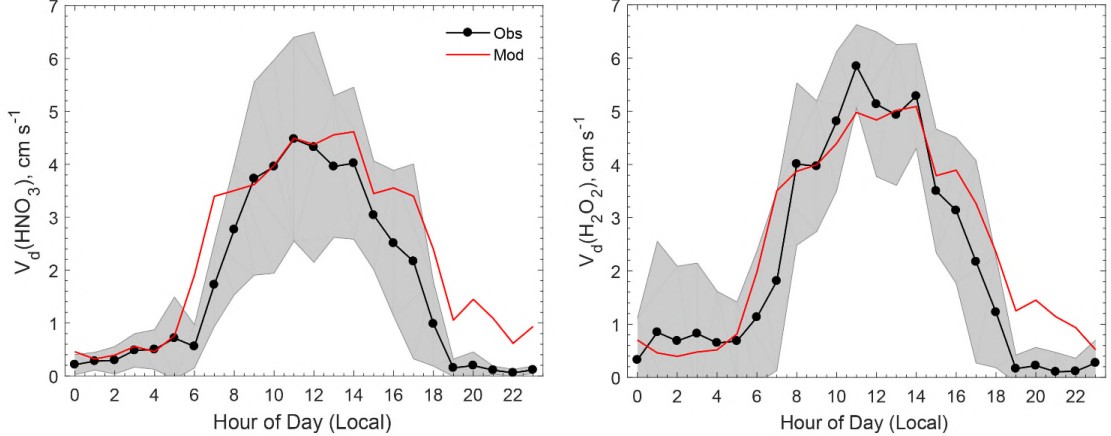

Figure 1. Comparison of the observed and modeled average diel variations of dry deposition velocities ($V_d$) for $HNO_3$ and $H_2O_2$. The shaded area indicates the standard deviation of the observations. The model assumes that surface resistances ($R_c$) for $HNO_3$ and $H_2O_2$ are zero.





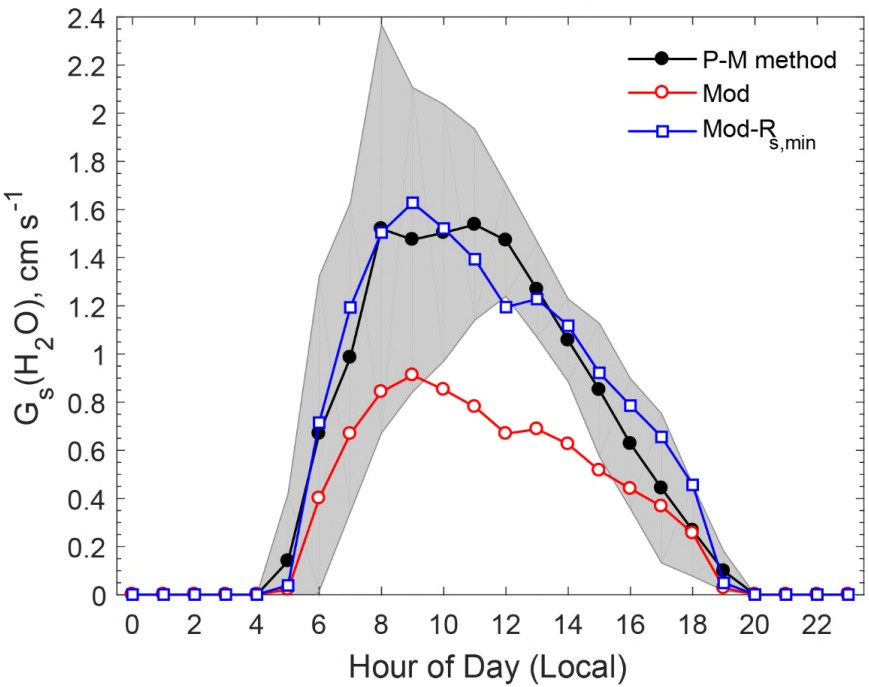

Figure 2. Comparison of observation-based and modeled averaged diel variations of stomatal conductance ($G_s$) for water vapor. The shaded area indicates the standard deviation of the observation-based $G_s(H_2O)$ estimated by the P-M method. "Mod-$R_{s,min}$" refers to a model sensitivity test in which $R_{s,min}$ was reduced by 40%.



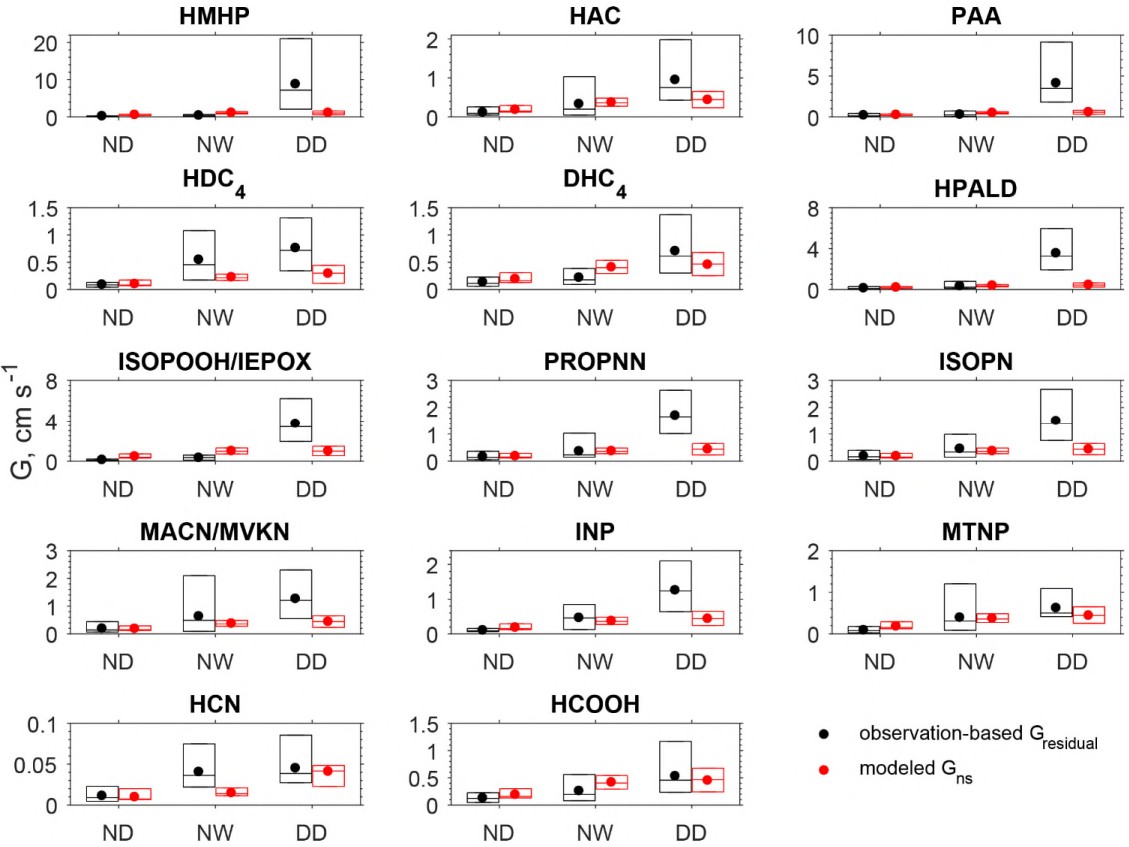

Figure 3. Box plot of the observation-based residual conductance ($G_{residual}$) and the modeled non-stomatal conductance ($G_{ns}$) during nighttime dry period (ND, n=88), nighttime wet period (NW, n=40), and daytime dry period (DD, n=85). In each box, the central mark is the median, and the edges of the box are the 25th and 75th percentiles. The filled dots represent the arithmetical mean of data between 25th and 75th percentiles. Daytime is 09:00-17:00 (local time) and nighttime is 20:00-06:00 (local time). The wet surface conditions were determined in the model driven by the observations of relative humidity, precipitation rate, friction velocity, and temperature.



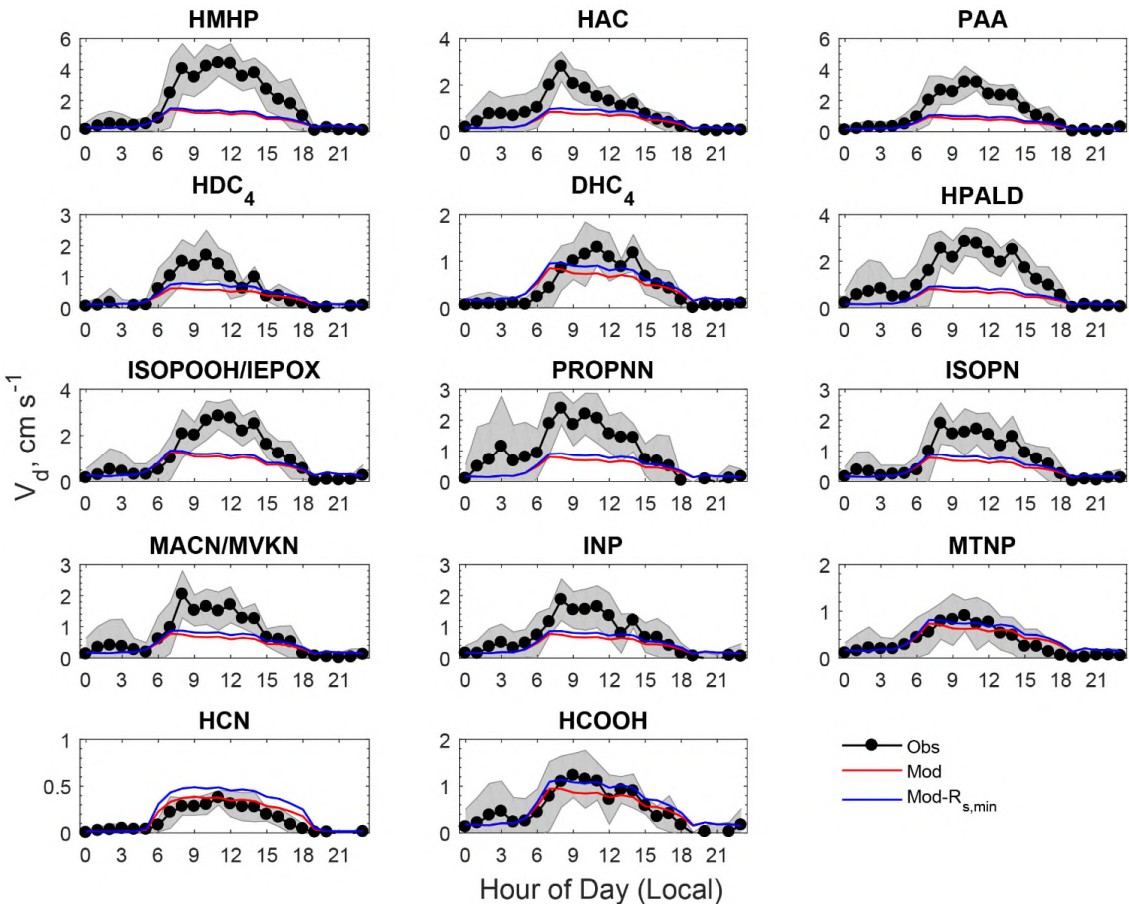

Figure 4. Comparison of averaged diel cycles of observed and modeled dry deposition velocities ($V_d$) of oVOCs and HCN. The shaded area indicates the standard deviation of the observations. "Mod-$R_{s,min}$" refers to a sensitivity test in which $R_{s,min}$ was reduced by 40%.





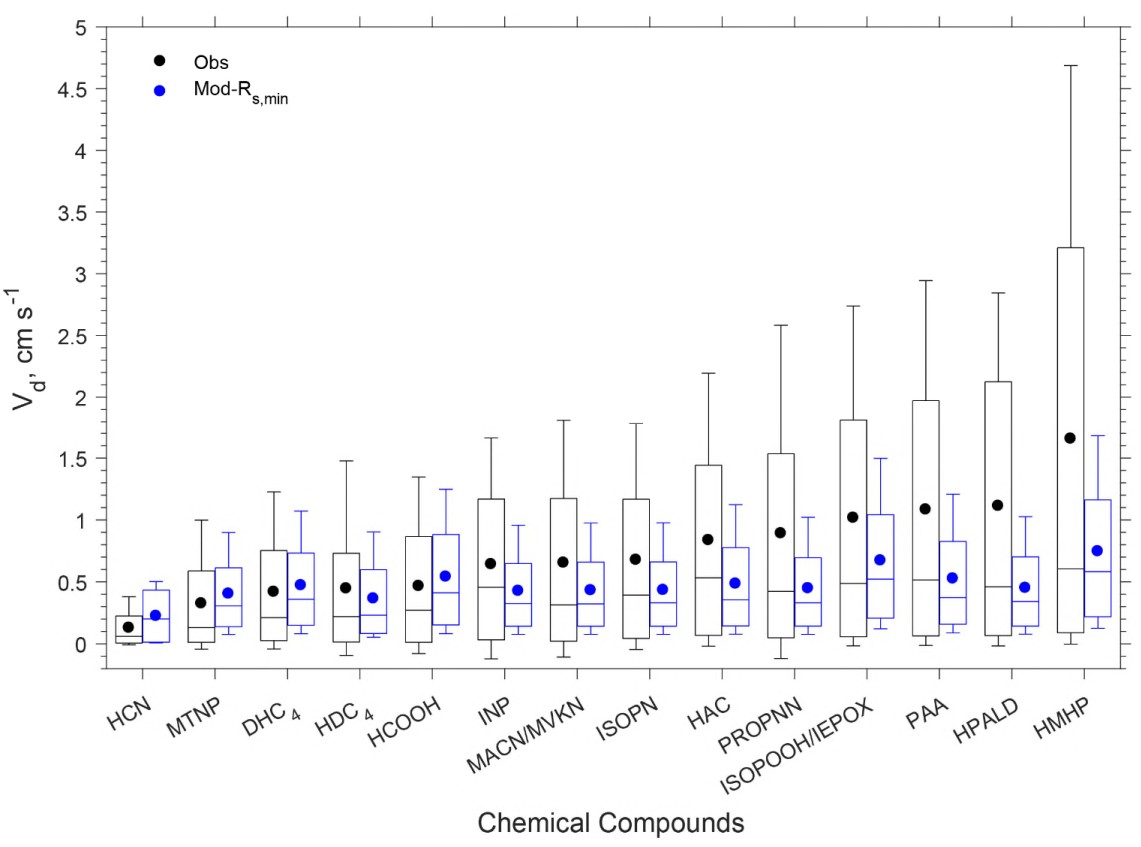

Figure 5. Box plot of observed and modeled hourly dry deposition velocities ($V_d$) of oVOCs and HCN. In each box, the central mark is the median, the edges of the box are the 25th and 75th percentiles, and the whiskers extend to the 10th and 90th percentiles. The filled dots represent the arithmetical mean of all the data. "Mod-$R_{s,min}$" refers to a sensitivity test in which $R_{s,min}$ was reduced by 40%.



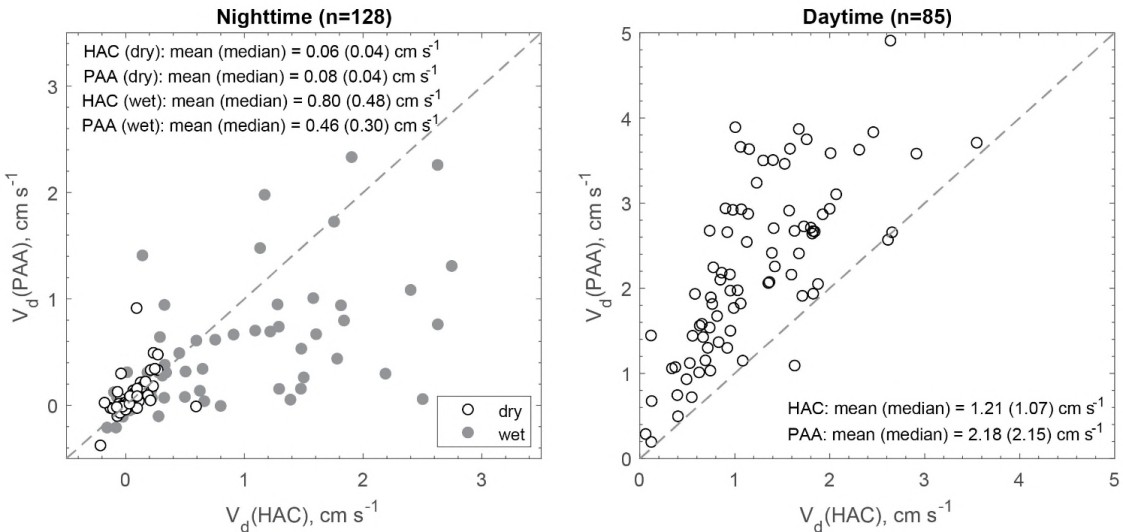

Figure 6. Scatter plot of the measured dry deposition velocities ($V_d$) for hydroxyacetone (HAC) and peroxyacetic acid (PAA) during nighttime (20:00-06:00, local time) and daytime (09:00-17:00, local time). The shaded (white) cycles correspond to the wet (dry) surface conditions.