# Peer review of "Extension of a gaseous dry deposition algorithm to oxidized volatile organic compounds and hydrogen cyanide for application in chemistry transport models"

_Geoscientific Model Development, 2021_

## Author Comment (AC1)

```fortran
      PROGRAM DRYDEP

      IMPLICIT REAL(A-H,O-Z),INTEGER(I-N)
      PARAMETER (NG=45)  ! number of gaseous species
      PARAMETER (NLUC=26)   ! number of land use category
      REAL VDG(NLUC,NG)
      REAL Vdmax(NLUC,NG),Rns(NLUC,NG),Dratio(NG)
      INTEGER KDAY(12), KDAY1(12)
      DATA KDAY/0,31,59,90,120,151,181,212,243,273,304,334/
      DATA KDAY1/31,28,31,30,31,30,31,31,30,31,30,31/
      character*500 line
      real Ra(NLUC),RST(NLUC),Ustar(NLUC), LAI_F(NLUC), LAI(26,15)

C
C  calculate Vd for one site each time
C

C  initialize site-dependent information
      GLAT=32.90289*3.14/180.    ! latitude for the site location, change accoridng
to your site
      Z2  = 22.

      OPEN(55, file="MET.dat")
      read(55, '(A500)', ERR=475) line

475   continue

500   read(55, '(I5,3I3,12F17.10)', END=525) IYR, IMO, ID, IH,
     &  T2, Ts, U2, RH, SRAD, PREC, P0, SD, FCLD, RMOL, UstarObs,Wetness

C  create output file
      OPEN (22,file='VD.dat')
      OPEN (33,file='METOUT.dat')
      OPEN (44,file='Res.dat')
      OPEN (11,file='Vdmax.dat')

C  find Julian day
      JDAY=KDAY(IMO)+ID
      if (IMO.GT.2.and.MOD(IYR,4).EQ.0 ) JDAY=JDAY+1.

C -- Calculate solar zenith angle
      hour=real(IH)
      DECLIN=ASIN(SIN(23.5*3.14159/180.)*
     &        SIN((JDAY-81.)*2.*3.14159/365.))
      SHORT1=SIN(GLAT)*SIN(DECLIN)
      SHORT2=COS(GLAT)*COS(DECLIN)
      COSZE=(HOUR-12)*3.14159/12.
      COSZEN=SHORT1+SHORT2*COS(COSZE)
      COSZEN= amax1(0.,COSZEN)
```

```fortran
C Call GasVd to calculate Vd for 45 gaseous species

      call GasVd (Z2, u2, sd, t2, ts, srad, rh, fcld, prec,
     &            COSZEN, P0, jday, RMOL, UstarObs,Wetness,
     &            VDG, Ra, Ustar, Vdmax, RST,Rns,Dratio)

      write(22,'(I4,3I3,90F10.5)') IYR,IMO,ID,IH,
     &    (VDG(4,J)*100,J=1,NG), (VDG(7,J)*100,J=1,NG)  ! m/s -> cm/s
      write(11,'(I4,3I3,45F10.5)') IYR,IMO,ID,IH,
     &    (Vdmax(4,J)*100,J=1,NG) ! m/s -> cm/s

      write(33,'(I5,3I3,12F17.10)') IYR, IMO, ID, IH,
     &    T2,Ts,U2,RH, SRAD, PREC, P0, SD, FCLD, RMOL,UstarObs,Wetness
      write(44,'(I5,3I3,3F17.10,90F10.5,45F7.4)') IYR,IMO,ID,IH, Ra(4),
     &    RST(4), RST(7),(100./Rns(4,J),J=1,NG),(100./Rns(7,J),J=1,NG),
     &    (Dratio(J),J=1,NG)

      goto 500
525   continue

      close(11)
      close(22)
      close(33)
      close(44)

      STOP "!!!DONE!!!"

      END

      SUBROUTINE init1 (V, K1)
      real V(K1)
      do i=1,K1
        V(i)=0.
      end do
      return
      end

      SUBROUTINE init2 (V, K1, K2)
      real V(K1, K2)
      do i=1,K1
        do j=1,K2
          V(i,j)=0.
        end do
      end do
      return
      end

      real function amin1 (x,y)
        amin1=x
        if (y.lt.x) amin1=y
```

```
      return
      end

      real function amax1 (x,y)
        amax1=x
        if (y.gt.x) amax1=y
      return
      end

      SUBROUTINE GasVd ( z2, u2, sd, t2, ts, srad, rh, fcld, prec,
     &                   COSZEN, pmb, iday, RMOL, UstarObs,Wetness,
     &                   VDG, Ra, Ustar, Vdmax, RST, Rns,Dratio)
C
C    PURPOSE: Calculate dry deposition velocities for 45 gas species
C              including 31 species listed in Table 6 of Zhang et al. (2003) and
14 additional
C              oVOC species listed in Table 1 of Wu et al. (2021)
C
C    References:
C      Zhang et al., 2003. A revised parameterization for gaseous dry
C      deposition in air-quality model. Atmos. Chem. Phys., 3, 2067-2082,
C      https://doi.org/10.5194/acp-3-2067-2003
C
C      Wu et al., 2021. Extension of a gaseous dry deposition algorithm
C      to oxidized volatile organic compounds and hydrogen cyanide for
C      application in chemistry transport models. Geosci. Model Dev.
C      Discuss., 5:1-31, https://doi.org/10.5194/gmd-2021-41
C
C                        leiming.zhang@canada.ca
C----------------------------------------------------------------------
C        KEY   VARIABLES
C----------------------------------------------------------------------
C  alpha    | Scaling factor based on SO2 (no unit)
C  beta     | Scaling factor based on O3  (no unit)
C  brs      | Constant for stomatal resistance(W/m2)
C  bvpd     | Constant for water vapor pressure deficit  (kPa^-1)

C  coszen   | Cosine of solar zenith angle
C  fcld     | Cloud fraction (0.0-1.0)
C  fland    | fraction of Land types (%)
C  fsun     | fraction of sunlit leaves (0.0-LAI)
C  iday     | Julian day
C  lai      | Leaf area index  (no unit)
C  luc      | No. of land use category (26)
C  mw       | molecular weight for gaseous species
C           |    (g/mol)
C  ng       | No. of gaseous species dry deposited
C  pardir   | visible beam radiation (W/m2)
C  pardif   | diffuse visible radiation (W/m2)
C  pmb      | Surface pressure (mb)
```

```
C  prec       | hourly precipitation amount (mm/hour)
C  psi1       | Constant for leaf water potential(Mpa)
C  psi2       | Constant for leaf water potential(Mpa)
C  fsnow      | Snow fraction  (0.0-1.0)
C  ra         | Aerodynamic resistance (s/m)
C  rac        | IN-canopy aerodynamic resistance (s/m)
C  rb         | quasi-laminar resistance (s/m)
C  rc         | total surface resistance (s/m)
C  rcut       | cuticle resistance (s/m )
C  rcutdo     | Dry cuticle resistance for O3 (s/m)
C  rcutds     | Dry cuticle resistance for SO2 (s/m)
C  rcutwo     | Wet cuticle resistance for O3 (s/m)
C  rg         | Ground  resistance (s/m )
C  rgo        | Ground  resistance for O3 (s/m)
C  rgs        | Ground  resistance for SO2 (s/m)
C  rh         | relative humidity fraction (0.0-1.0)
C  rm         | mesophyll resistance (s/m)
C  rsmin      | minimum stomatal resistance (s/m)
C  rst        | Stomatal resistance (s/m)
C  sd         | Snow depth       (cm)
C  sdmax      | Maximum snow depth over which snow
C             |    fraction for leaves is 1 (cm )
C  srad       | Solar irradiance (w/m2)
C  t2         | Temperature  at first level (K)
C  ts         | Surface temperature (K)
C  tmin       | Minimum temperature for stomatal
C             |     opening (C)
C  tmax       | Maxmum temperature for stomatal
C             |     opening (C)
C  topt       | Optimum temperature for stomatal
C             |     opening (C)
C  u2         | wind speed at reference height z2(m/s)
C  ustar      | friction velocity (m/s)
C  VDF        | dry deposition velocity for one LUC
C  VDG        | gaseous dry deposition velocity (m/s)
C  wst        | fraction of stomatal closure under
C             |      wet conditions (0.0-0.5)
C  z0         | roughness length (m)
C----------------------------------------------------------------------

      IMPLICIT REAL(A-H,O-Z),INTEGER(I-N)
      PARAMETER (NG=45)        ! NUMBER OF GAS SPECIES DRY DEPOSITED
      PARAMETER (LUC=26)       ! NUMBER OF LAND-USE CATEGORIES

      REAL Ra(LUC),RST(LUC)
C
C  paramaters
C
      REAL Z01(LUC), Z02(LUC),  ustar(LUC)
```

```fortran
      REAL LAI(LUC,15),LAI_F(LUC)
C
C   paramaters for gaseous Vd submoudle
C
      REAL VDG(LUC,NG), ALPHA(NG),BETA(NG),RM(NG),MW(NG)
      REAL Rac1(LUC),   Rac2(LUC), RcutdO(LUC), RcutwO(LUC),
     &  RcutdS(LUC),   RgS(LUC),   RgO(LUC),  SDmax(LUC),
     &   Tmin(LUC),   Tmax(LUC),   TOPT(LUC),   BVPD(LUC),
     &   PSI1(LUC),   PSI2(LUC),  RSmin(LUC),    BRS(LUC)
      LOGICAL is_dew, is_rain
      REAL Vdmax(LUC,NG), Rns(LUC,NG),Dratio(NG)

C
C   external functions
C
      external amin1, amax1

C
C   Surface Roughness Length [m].
C   Z01 and Z02 are minimum and maximum z0 for each LUC.
C
      DATA    Z01              /
     & 0.0 ,  0.01,  0.0 ,  0.9 ,  2.0 ,
     & 0.4 ,  0.4 ,  2.5 ,  0.6 ,  0.2 ,
     & 0.05,  0.2 ,  0.04,  0.02,  0.02,
     & 0.02,  0.02,  0.02,  0.02,  0.05,
     & 1.0 ,  0.03,  0.1 ,  0.04,  0.6 ,
     & 0.6    /
      DATA    Z02              /
     & 0.0 ,  0.01,  0.0 ,  0.9 ,  2.0 ,
     & 0.9 ,  1.0 ,  2.5 ,  0.6 ,  0.2 ,
     & 0.2 ,  0.2 ,  0.04,  0.1 ,  0.1 ,
     & 0.1 ,  0.1 ,  0.1 ,  0.2 ,  0.05,
     & 1.0 ,  0.03,  0.1 ,  0.04,  0.9 ,
     & 0.9    /
C
C   In-canopy aerodynamic resistance  [s/m].
C   Rac1 and Rac2 are minimum and maximum Rac0 for each LUC.
C
      DATA    Rac1              /
     & 0   ,  0   ,  0   ,  100 ,  250 ,
     & 60  ,   60 ,  300 ,  100 ,  60  ,
     & 20  ,  40  ,  20  ,  10  ,  10  ,
     & 10  ,  10  ,  10  ,  10  ,  20  ,
     & 40  ,  0   ,  20  ,  0   ,  100 ,
     & 100    /
      DATA    Rac2              /
     & 0   ,  0   ,  0   ,  100 ,  250 ,
     & 100 ,  100 ,  300 ,  100 ,  60  ,
     & 60  ,  40  ,  20  ,  40  ,  40  ,
```

```
      &  40  ,  40  ,  50  ,  40  ,  20  ,
      &  40  ,  0   ,  20  ,  0   ,  100 ,
      &  100     /
C
C    Dry and wet cuticle resistance for O3  [s/m].
C
       DATA    RcutdO           /
      & -999 , -999 , -999 , 4000 , 6000 ,
      & 4000 , 6000 , 6000 , 8000 , 6000 ,
      & 5000 , 5000 , 4000 , 4000 , 4000 ,
      & 4000 , 4000 , 5000 , 5000 , 4000 ,
      & 6000 , 8000 , 5000 , -999 , 4000 ,
      & 4000     /
       DATA    RcutwO           /
      & -999 , -999 , -999 ,  200 ,  400 ,
      &  200 ,  400 ,  400 ,  400 ,  400 ,
      &  300 ,  300 ,  200 ,  200 ,  200 ,
      &  200 ,  200 ,  300 ,  300 ,  200 ,
      &  400 ,  400 ,  300 , -999 ,  200 ,
      &  200     /
C
C    Ground resistance for O3  [s/m].
C
       DATA    RgO              /
      & 2000 , 2000 , 2000 ,  200 ,  200 ,
      &  200 ,  200 ,  200 ,  200 ,  200 ,
      &  200 ,  200 ,  200 ,  200 ,  200 ,
      &  200 ,  200 ,  200 ,  200 ,  500 ,
      &  500 ,  500 ,  500 ,  500 ,  200 ,
      &  200     /
C
C    Dry cuticle resistance for SO2  [s/m].
C
       DATA    RcutdS           /
      & -999 , -999 , -999 , 2000 , 2500 ,
      & 2000 , 2500 , 2500 , 6000 , 2000 ,
      & 2000 , 2000 , 1000 , 1000 , 1500 ,
      & 1500 , 2000 , 2000 , 2000 , 2000 ,
      & 4000 , 2000 , 1500 , -999 , 2500 ,
      & 2500     /
C
C    Ground resistance for SO2  [s/m].
C
       DATA    RgS              /
      &   20 ,   70 ,  20  ,  200 ,  100 ,
      &  200 ,  200 ,  100 ,  300 ,  200 ,
      &  200 ,  200 ,  200 ,  200 ,  200 ,
      &   50 ,  200 ,  200 ,  200 ,   50 ,
      &  300 ,  300 ,   50 ,  700 ,  200 ,
      &  200     /
```

```
C
C   Stomatal resistance related parameters.
C   In sequence: rsmin, brs, tmin, tmax, topt, bvpd, psi1, psi2
C
      DATA    rsmin            /
     & -999 , -999 , -999 ,   250 ,   150 ,
     &  250 ,   150,   150 ,   250 ,   150 ,
     &  150 ,   250 ,   150 ,   100 ,   120 ,
     &  120 ,   120 ,   250 ,   125 ,   150 ,
     &  200 ,   150 ,   150 , -999 ,   150 ,
     &  150    /
      DATA    brs              /
     & -999 , -999 , -999 ,    44 ,    40 ,
     &   44 ,    43 ,    40 ,    44 ,    40 ,
     &   44 ,    44 ,    50 ,    20 ,    40 ,
     &   40 ,    50 ,    65 ,    65 ,    40 ,
     &   42 ,    25 ,    40 , -999 ,    44 ,
     &   43    /
      DATA    tmin             /
     & -999 , -999 , -999 ,    -5 ,     0 ,
     &   -5 ,     0 ,     0 ,     0 ,     0 ,
     &   -5 ,     0 ,     5 ,     5 ,     5 ,
     &    5 ,     5 ,     5 ,    10 ,     5 ,
     &    0 ,    -5 ,     0 , -999 ,    -3 ,
     &    0    /
      DATA    tmax             /
     & -999 , -999 , -999 ,    40 ,    45 ,
     &   40 ,    45 ,    45 ,    45 ,    45 ,
     &   40 ,    45 ,    40 ,    45 ,    45 ,
     &   45 ,    45 ,    45 ,    45 ,    45 ,
     &   45 ,    40 ,    45 , -999 ,    42 ,
     &   45    /
      DATA    topt             /
     & -999 , -999 , -999 ,    15 ,    30 ,
     &   15 ,    27 ,    30 ,    25 ,    30 ,
     &   15 ,    25 ,    30 ,    25 ,    27 ,
     &   27 ,    25 ,    25 ,    30 ,    25 ,
     &   22 ,    20 ,    20 , -999 ,    21 ,
     &   25    /
      DATA    bvpd             /
     & -999 , -999 , -999 ,  0.31,  0.27,
     &  0.31,  0.36,  0.27,  0.31,  0.27,
     &  0.27,  0.27,  0.0 ,  0.0 ,  0.0 ,
     &  0.0 ,  0.0 ,  0.0 ,  0.0 ,  0.0 ,
     &  0.31,  0.24,  0.27, -999 ,  0.34,
     &  0.31    /
      DATA    psi1             /
     & -999 , -999 , -999 , -2.0 , -1.0 ,
     & -2.0 , -1.9 , -1.0 , -1.0 , -2.0 ,
     & -2.0 , -2.0 , -1.5 , -1.5 , -1.5 ,
```

```
      & -1.5 , -1.5 , -1.5 , -1.5 , -1.5 ,
      & -1.5 ,    0 , -1.5 , -999 , -2.0 ,
      & -2.0    /
       DATA   psi2               /
      & -999 , -999 , -999 , -2.5 , -5.0 ,
      & -2.5 , -2.5 , -5.0 , -4.0 , -4.0 ,
      & -4.0 , -3.5 , -2.5 , -2.5 , -2.5 ,
      & -2.5 , -2.5 , -2.5 , -2.5 , -2.5 ,
      & -3.0 , -1.5 , -2.5 , -999 , -2.5 ,
      & -3.0    /
C
C Leaf area index at the beginning of each month (im=1,13),
C  minimum LAI (im=14) and maximum LAI (im=15).
C Values of LAI are from GEM, provided by Stephane Belair and Judy St-James,
C with modifications for urban.
C
       DATA (LAI(6,im), im = 1, 15)/
      & 0.1  , 0.1  , 0.5  , 1.0  , 2.0  ,
      & 4.7  , 4.7  , 5.0  , 4.0  , 2.0  ,
      & 1.0  , 0.1  , 0.1  , 0.1  , 5.0  /
       DATA (LAI(7,im), im = 1, 15)/
      & 0.1  , 0.1  , 0.5  , 1.0  , 2.0  ,
      & 4.7  , 4.7  , 5.0  , 4.0  , 2.0  ,
      & 1.0  , 0.1  , 0.1  , 0.1  , 5.0  /
       DATA (LAI(11,im), im = 1, 15)/
      & 0.5  , 0.5  , 1.0  , 1.0  , 1.5  ,
      & 2.0  , 3.0  , 3.0  , 2.0  , 1.5  ,
      & 1.0  , 0.5  , 0.5  , 0.5  , 3.0  /
       DATA (LAI(14,im), im = 1, 15)/
      & 0.5  , 0.5  , 0.5  , 0.5  , 0.5  ,
      & 0.5  , 1.0  , 2.0  , 2.0  , 1.5  ,
      & 1.0  , 1.0  , 0.5  , 0.5  , 2.0  /
       DATA (LAI(15,im), im = 1, 15)/
      & 0.1  , 0.1  , 0.1  , 0.5  , 1.0  ,
      & 2.0  , 3.0  , 3.5  , 4.0  , 0.1  ,
      & 0.1  , 0.1  , 0.1  , 0.1  , 4.0  /
       DATA (LAI(16,im), im = 1, 15)/
      & 0.1  , 0.1  , 0.1  , 0.5  , 1.0  ,
      & 2.5  , 4.0  , 5.0  , 6.0  , 0.1  ,
      & 0.1  , 0.1  , 0.1  , 0.1  , 6.0  /
       DATA (LAI(17,im), im = 1, 15)/
      & 0.1  , 0.1  , 0.1  , 0.5  , 1.0  ,
      & 3.0  , 4.0  , 4.5  , 5.0  , 0.1  ,
      & 0.1  , 0.1  , 0.1  , 0.1  , 5.0  /
       DATA (LAI(18,im), im = 1, 15)/
      & 0.1  , 0.1  , 0.1  , 0.5  , 1.0  ,
      & 2.0  , 3.0  , 3.5  , 4.0  , 0.1  ,
      & 0.1  , 0.1  , 0.1  , 0.1  , 4.0  /
       DATA (LAI(19,im), im = 1, 15)/
      & 0.1  , 0.1  , 0.1  , 0.5  , 1.0  ,
```

```
      & 3.0  , 4.0  , 4.5  , 5.0  , 0.1  ,
      & 0.1  , 0.1  , 0.1  , 0.1  , 5.0  /
       DATA (LAI(21,im), im = 1, 15)/
      & 0.1  , 0.1  , 0.1  , 0.1  , 0.5  ,
      & 1.0  , 1.0  , 1.0  , 1.0  , 1.0  ,
      & 0.4  , 0.1  , 0.1  , 0.1  , 1.0  /
       DATA (LAI(22,im), im = 1, 15)/
      & 1.0  , 1.0  , 0.5  , 0.1  , 0.1  ,
      & 0.1  , 0.1  , 1.0  , 2.0  , 1.5  ,
      & 1.5  , 1.0  , 1.0  , 0.1  , 2.0  /
       DATA (LAI(25,im), im = 1, 15)/
      & 3.0  , 3.0  , 3.0  , 4.0  , 4.5  ,
      & 5.0  , 5.0  , 5.0  , 4.0  , 3.0  ,
      & 3.0  , 3.0  , 3.0  , 3.0  , 5.0  /
       DATA (LAI(26,im), im = 1, 15)/
      & 3.0  , 3.0  , 3.0  , 4.0  , 4.5  ,
      & 5.0  , 5.0  , 5.0  , 4.0  , 3.0  ,
      & 3.0  , 3.0  , 3.0  , 3.0  , 5.0  /
C
C  Gas Properties (Mesophyll resistance RM, scaling factors ALPHA and BETA,
C  and molecular weight) for a total of 45 species in this sequence:
C
C     1  SO2
C     2  H2SO4
C     3  NO2
C     4  O3
C     5  H2O2
C     6  HNO3
C     7  HONO
C     8  HNO4
C     9  NH3
C    10 PAN
C    11 PPN
C    12 APAN
C    13 MPAN
C    14 HCHO
C    15 MCHO
C    16 PALD
C    17 C4A
C    18 C7A
C    19 ACHO
C    20 MVK
C    21 MACR
C    22 MGLY
C    23 MOH
C    24 ETOH
C    25 POH
C    26 CRES
C    27 FORM
C    28 ACAC
```

```
C     29 ROOH
C     30 ONIT
C     31 INIT
C     32 HCN
C     33 HMHP
C     34 HAC
C     35 PAA
C     36 HDC4
C     37 DHC4
C     38 HPALD
C     39 ISOPOOH
C     40 IEPOX
C     41 PROPNN
C     42 ISOPN
C     43 MACN/MVKN
C     44 INP
C     45 MTNP

      DATA    RM              /
     & 0.    , 0.    , 0.    , 0.   , 0.   ,
     & 0.    , 0.    , 0.    , 0.   , 0.   ,
     & 0.    , 0.    , 0.    , 0.   , 100.,
     & 100.  , 100.  , 100.  , 100., 0.   ,
     & 100.  , 0.    , 0.    , 0.   , 0.   ,
     & 0.    , 0.    , 0.    , 0.   , 100.,
     & 100.  , 100.  , 0.    , 0.   , 0.   ,
     & 0.    , 0.    , 0.    , 0.   , 0.   ,
     & 0.    , 0.    , 0.    , 0.   , 0./

      DATA    ALPHA           /
     & 1.    , 1.    , 0.    , 0.   , 1.   ,
     & 10.   , 2.    , 5.    , 1.   , 0.   ,
     & 0.    , 0.    , 0.    , 0.8  , 0.   ,
     & 0.    , 0.    , 0.    , 0.   , 0.   ,
     & 0.    , 0.01  , 0.6   , 0.6  , 0.4  ,
     & 0.01  , 2.    , 1.5   , 0.1  , 0.   ,
     & 0.    , 0.    , 5.    , 1.5  , 2.   ,
     & 1.    , 2.    , 1.5   , 5.   , 5.   ,
     & 1.5   , 1.5   , 1.5   , 1.5  , 1.5  /

      DATA    BETA            /
     & 0.    , 1.    , 0.8   , 1.   , 1.   ,
     & 10.   , 2.    , 5.    , 0.0  , 0.6  ,
     & 0.6   , 0.8   , 0.3   , 0.2  , 0.05 ,
     & 0.05  , 0.05  , 0.05  , 0.05 , 0.05 ,
     & 0.05  , 0.    , 0.1   , 0.   , 0.   ,
     & 0.    , 0.2   , 0.    , 0.8  , 0.5  ,
     & 0.5   , 0.1   , 1.    , 1.   , 1.   ,
     & 0.2   , 0.2   , 1.    , 0.2  , 0.2  ,
     & 1.    , 1.    , 1.    , 1.   , 1./
```

```
      DATA    MW            /
     & 64.  , 98.  , 46.  ,  48. ,  34. ,
     & 63.  , 47.  , 79.  ,  17. ,  121.,
     & 135. , 183. , 147. ,  30. ,  44. ,
     & 58.  , 72.  , 128. ,  106.,  70. ,
     & 70.  , 72.  , 32.  ,  46. ,  60. ,
     & 104. , 46.  , 60.  ,  48. ,  77. ,
     & 147. , 27.  , 64.  ,  74. ,  76. ,
     & 102. , 104. , 116. ,  118.,  118.,
     & 119. , 147. , 149. ,  163.,  231./
C
C  Maximum snow depth over which snow fraction for leaves is 1.0
C  Snow fraction for ground is treated 2 times of that for leaves
C
      DATA SDMAX  /
     &               9999. , 1.0   , 9999. ,  200. ,  400. ,
     &                200. , 200.  ,  400. ,  200. ,   50. ,
     &                 50. , 50.   ,    5. ,   20. ,   10. ,
     &                 10. , 10.   ,   10. ,   10. ,   10. ,
     &                 50. ,  2.   ,   10. ,    2. ,  200. ,
     &                200.          /

C ---   parameters for air dynamic properties
      DATA ROAROW/1.19/

C
C --- Define the function for saturation vapor pressure (mb)
C
      ES(TEMP) = 6.108*EXP(17.27*(TEMP - 273.16)/(TEMP - 35.86))
C
C   Some constants
C
      dair=0.369*29.+6.29
      dh2o=0.369*18.+6.29

C   Initialize Leaf Area Index for LUC with constant LAI values
      DO im=1, 15
      LAI(1,im)=0.
      LAI(2,im)=0.
      LAI(3,im)=0.
      LAI(4,im)=4.7
      LAI(5,im)=6.
      LAI(8,im)=6.
      LAI(9,im)=4.
      LAI(10,im)=3.
      LAI(12,im)=3.
      LAI(13,im)=1.
      LAI(20,im)=1.
      LAI(23,im)=4.
```

```fortran
      LAI(24,im)=0.
      END DO

      VDG=0.
      Vdmax=0.
      Rns=0.

C
C Loop 200 for LUC
C
      DO 200 I=1,LUC

C
C interpolate LAI
C
         IM = INT(iday / 30.5 ) + 1
         iday_M =iday - INT((IM-1)*30.5+0.5)
         IF (iday_M.EQ.0) THEN
         IM=IM-1
         iday_M =iday - (IM-1)*30.5
         END IF

         LAI_F(I)  = LAI(I,IM)
     &          + iday_M / 30.5 * (LAI(I,IM+1)-LAI(I,IM))

      Z0_F  = (0.23-LAI_F(I)**0.25/10-(2-1)/67.)*10.
      DDD   = (0.05+LAI_F(I)**0.20/2.+(2-1)/20.)*10.
      ZL=(Z2-DDD)*RMOL
      USTAR(I)= UstarObs

C
C     Aerodynamic resistance above canopy
C
      IF(ZL.GE.0.) THEN
          Ra(I)=(.74*ALOG((Z2-DDD)/Z0_F)+4.7*ZL)/0.4/USTAR(I)
      ELSE
          Ra(I)=0.74/0.4/USTAR(I)*(ALOG((Z2-DDD)/Z0_F)-
     &          2*ALOG((1+SQRT(1-9.*ZL))*0.5))
      ENDIF

      Ra(I)=amax1(Ra(I),1.0)

      if (I.EQ.1.OR.I.EQ.3) THEN
          Ra(I)=amin1(Ra(I),2000.)
      else
          Ra(I)=amin1(Ra(I),1000.)
      end if

C
C --- STOMATAL RESISTANCE FOR WATER VAPOR ONLY. STEPS FOR CALCULATING:
```

```
C      1. Calculate direct and diffuse PAR from solar radiation
C      2. Calculate sunlit and shaded leaf area, PAR for sunlit and shaded leafs
C      3. Calculate stomatal conductance
C      4. Calculate stomatal resistance for water vapor
C
C
C --  Only calculate stomatal resistance if there is solar radiation,
C      leaf area index is not zero, and within reasonable temperature range
C
       IF ( SRAD.GE.0.1               .AND.
     &      TS.LT.(Tmax(I)+273.15)   .AND.
     &      TS.GT.(Tmin(I)+273.15)   .AND.
     &      LAI_F(I).GT.0.001        .AND.
     &      COSZEN.GT.0.001          ) THEN

C --  Calculate direct and diffuse PAR from solar radiation and solar zenith angle

       RDU=600.*EXP(-0.185/COSZEN)*COSZEN
       RDV=0.4*(600.-RDU)*COSZEN
       WW=-ALOG(COSZEN)/2.302585
       WW=-1.195+0.4459*WW-0.0345*WW**2
       WW=1320*10**WW
       RDM=(720.*EXP(-0.06/COSZEN)-WW)*COSZEN
       RDN=0.6*(720-RDM-WW)*COSZEN
       RV=amax1(0.1,RDU+RDV)
       RN=amax1(0.01,RDM+RDN)
       RATIO=amin1(0.9,SRAD/(RV+RN))
       SV=RATIO*RV                              ! Total PAR
       FV=amin1(0.99, (0.9-RATIO)/0.7)
       FV=amax1(0.01,RDU/RV*(1.0-FV**0.6667))  !fraction of PAR in the direct beam
       PARDIR=FV*SV                             ! PAR from direct radiation
       PARDIF=SV-PARDIR                         ! PAR from diffuse radiation
c
C -- Calculate sunlit and shaded leaf area, PAR for sunlit and shaded leaves
C
       IF (LAI_F(I).GT.2.5.AND.SRAD.GT.200.) THEN
       PSHAD=PARDIF*EXP(-0.5*LAI_F(I)**0.8)
     & +0.07*PARDIR*(1.1-0.1*LAI_F(I))*EXP(-COSZEN)
       PSUN=PARDIR**0.8*.5/COSZEN+PSHAD
       ELSE
       PSHAD=PARDIF*EXP(-0.5*LAI_F(I)**0.7)
     & +0.07*PARDIR*(1.1-0.1*LAI_F(I))*EXP(-COSZEN)
       PSUN=PARDIR*.5/COSZEN+PSHAD
       END IF
       RSHAD=RSmin(I)+BRS(I)*RSMIN(I)/PSHAD
       RSUN=RSmin(I)+BRS(I)*RSMIN(I)/PSUN
       GSHAD=1./RSHAD
       GSUN=1./RSUN
       FSUN=2*COSZEN*(1.-EXP(-0.5*LAI_F(I)/COSZEN))  ! Sunlit leaf area
       FSHAD=LAI_F(I)-FSUN                       ! Shaded leaf area
```

```fortran
C -- Stomatal conductance before including effects of temperature,
C                      vapor pressure defict and water stress.

      GSPAR=FSUN*GSUN+FSHAD*GSHAD

C --   function for temperature effect
      T=TS-273.15
      BT=(Tmax(I)-TOPT(I))/(TOPT(I)-Tmin(I))
      GT=(Tmax(I)-T)/(TMAX(I)-TOPT(I))
      GT=GT**BT
      GT=GT*(T-Tmin(I))/(TOPT(I)-TMIN(I))
C --   function for vapor pressure deficit
      D0= ES(TS)*(1.- RH*1.05)/10.           !kPa
      GD=1.-BVPD(I)*D0
C --   function for water stress
      PSI=(-0.72-0.0013*SRAD)
      GW=(PSI-PSI2(I))/(PSI1(I)-PSI2(I))
      IF (GW.GT.1.0) GW=1.0
      IF (GW.LT.0.1) GW=0.1
      IF (GD.GT.1.0) GD=1.0
      IF (GD.LT.0.1) GD=0.1
C --   Stomatal resistance for water vapor
      RST(I)=1.0/(GSPAR*GT*GD*GW)

      ELSE
      RST(I)=99999.9
      END IF
C
c    Decide if dew or rain occurs.
C
      IF (FCLD.LT.0.25) THEN
        Coedew=0.3
      ELSE  IF (FCLD.GE.0.25.AND.FCLD.LT.0.75) THEN
        Coedew=0.2
      ELSE
        Coedew=0.1
      END IF
        DQ=0.622/1000. * ES(TS)*(1.- RH)*1000.    ! unit g/kg
        DQ=amax1(0.0001,DQ)
        USMIN=1.5/DQ*Coedew

      is_rain = .false.
      is_dew  = .false.
      IF (T2.GT.273.15) THEN
        if (PREC.GT.0.20) then
           is_rain = .true.
        elseif (Wetness.GT.0.8) then
           is_dew  = .true.
        endif
```

```
      ENDIF

C
C   Decide fraction of stomatal blocking due to wet conditions
C
      Wst=0.
      if ((is_dew.or.is_rain).and.SRAD.GT.200.) then
      Wst=(SRAD-200.)/800.
      Wst=amin1(Wst, 0.5)
      end if
C
C -- In-canopy aerodynamic resistance
C
          Rac = Rac1(I)+(LAI_F(I)-LAI(I,14))/(LAI(I,15)-LAI(I,14)+1.E-10)
     &              *(Rac2(I)-Rac1(I))
          Rac = Rac*LAI_F(I)**0.25/USTAR(I)/USTAR(I)
C
C -- Ground resistance for O3
C
      IF (I.GE.4.AND.TS.LT.272.15) THEN
        RgO_F = amin1( RgO(I)*2., RgO(I) * exp(0.2*(272.15-TS)))
      ELSE
        RgO_F = RgO(I)
      END IF
C
C -- Ground resistance for SO2
C
      IF (I.EQ.2) THEN
        RgS_F = AMIN1(RgS(I)*(275.15-TS), 500.)
        RgS_F = AMAX1(RgS(I), 100.)
      ELSE IF (I.GE.4.AND.is_rain) THEN
        RgS_F = 50.
      ELSE IF (I.GE.4.AND.is_dew) THEN
        RgS_F = 100.
      ELSE IF (I.GE.4.AND.TS.LT.272.15) THEN
        RgS_F = amin1( RgS(I)*2., RgS(I) * exp(0.2*(272.15-TS)))
      ELSE
        RgS_F =  RgS(I)
      END IF
C
C -- Cuticle resistance for O3 AND SO2
C
      IF (RcutdO(I).LE.-1) THEN
        RcutO_F = 1.E25
        RcutS_F = 1.E25
      ELSE IF (is_rain) THEN
        RcutO_F = RcutwO(I)/LAI_F(I)**0.5/USTAR(I)
        RcutS_F = 50./LAI_F(I)**0.5/USTAR(I)
        RcutS_F = MAX (RcutS_F, 20.)
      ELSE IF (is_dew) THEN
```

```
           RcutO_F = RcutwO(I)/LAI_F(I)**0.5/USTAR(I)
           RcutS_F = 100./LAI_F(I)**0.5/USTAR(I)
           RcutS_F = MAX (RcutS_F, 20.)
         ELSE IF (TS.LT.272.15) THEN
           RcutO_F = RcutdO(I)/exp(3.*RH)/LAI_F(I)**0.25/USTAR(I)
           RcutS_F = RcutdS(I)/exp(3.*RH)/LAI_F(I)**0.25/USTAR(I)
           RcutO_F = amin1( RcutO_F*2., RcutO_F * exp(0.2*(272.15-TS)))
           RcutS_F = amin1( RcutS_F*2., RcutS_F * exp(0.2*(272.15-TS)))
           RcutO_F = MAX (RcutO_F,100.)
           RcutS_F = MAX (RcutS_F,100.)
         ELSE
           RcutO_F = RcutdO(I)/exp(3.*RH)/LAI_F(I)**0.25/USTAR(I)
           RcutS_F = RcutdS(I)/exp(3.*RH)/LAI_F(I)**0.25/USTAR(I)
           RcutO_F = MAX (RcutO_F,100.)
           RcutS_F = MAX (RcutS_F,100.)
         END IF
C
C If snow occurs, Rg and Rcut are adjusted by snow cover fraction
C
           fsnow= sd/sdmax(i)
           fsnow= amin1(1.0, fsnow)    !snow cover fraction for leaves
         If (fsnow.GT.0.0001.and.I.GE.4) THEN
           RsnowS= AMIN1(70.*(275.15-TS), 500.)
           RsnowS= AMAX1(RSnowS, 100.)
           RcutS_F=1.0/((1.-fsnow)/RcutS_F+fsnow/RsnowS)
           RcutO_F=1.0/((1.-fsnow)/RcutO_F+fsnow/2000.)
           fsnow= amin1(1.0, fsnow*2.)    !snow cover fraction for ground
           RgS_F=1.0/((1.-fsnow)/RgS_F+fsnow/RsnowS)
           RgO_F=1.0/((1.-fsnow)/RgO_F+fsnow/2000.)
         END IF

C
C Loop 100 for gas species
C
         DO 100 J=1,NG
C
C -- Calculate diffusivity for each gas species
C
           dgas=0.369*MW(J)+6.29
           DI=0.001*TS**1.75*SQRT((29.+MW(J)))/MW(J)/29.)
           DI=DI/1.0/(dair**0.3333+dgas**0.3333)**2
           VI=145.8*1.E-4*(TS*0.5+T2*0.5)**1.5/
     &                   (TS*0.5+T2*0.5+110.4)
           VI=VI/ROAROW
C
C -- Calculate quasi-laminar resistance
C
           Rb  =5./USTAR(I)*(VI/DI)**.666667
C
C -- Calculate stomatal resistance for each species from the ratio of
```

```fortran
C         diffusity of water vapor to the gas species
C
          DVh2o=0.001*TS**1.75*SQRT((29.+18.)/29./18.)
          DVh2o=DVh2o/(dair**0.3333+dh2o**0.3333)**2

          RS=RST(I)*DVh2o/DI+RM(J)
C
C -- Scale cuticle and ground resistances for each species
C
          Rcut = 1./(ALPHA(J)/RcutS_F+BETA(J)/RcutO_F)
          Rg = 1./(ALPHA(J)/RgS_F+BETA(J)/RgO_F)
C
C -- Calculate total surface resistance
C
          Rc = (1.-Wst)/Rs+1./(Rac+Rg)+1./Rcut
          Rc=amax1(0.,1./Rc)

C
C -- Deposition velocity
C
          VDG(I,J) = 1./(RA(I)+RB+RC)
          Vdmax(I,J) = 1./(RA(I)+RB)
          Rns(I,J) = 1./(1./(Rac+Rg)+1./Rcut)
          Dratio(J) = DVh2o/DI

100   CONTINUE   ! end of gaseous species
200   CONTINUE   ! end of LUC

      RETURN

      END SUBROUTINE GasVd
```

---

## Author Comment (AC4)

**Response to Reviewer #1**

We greatly appreciate the reviewer for providing valuable comments on our manuscript, which have helped us improve the paper quality. We have addressed all of the comments carefully as detailed below. The original comments are in black and our replies are in blue.

Summary:

In "Extension of a gaseous dry deposition algorithm to oxidized volatile organic compounds and hydrogen cyanide for application in chemistry transport models", Wu et al. describe an extension of an existing dry deposition algorithm to 12 additional oxidized VOCs and evaluation of the model against field data. The effort shows that some oVOCs are well-represented by this formulation, but others severely underestimate the observed deposition rates, suggesting a second sink is also important that the authors suggest is chemical reactivity. Overall, the important content is included, but the manuscript would be improved with a reorganization to introduce important background information earlier on. Specific comments aim to improve this and other areas of the work.

Major comments:

As an overall comment, the manuscript would benefit from reorganization in ways that present relevant background information earlier in the introduction and methods, and not waiting in some cases to present this information in the results and discussion. An additional section heading after 3.2 could help indicate that the discussion has shifted from evaluation of modeled deposition velocities to the role of other loss mechanisms, namely chemical reactions. The comments below give more specific examples for this organization along with other notes.

We have added some additional materials in Abstract and Introduction based on reviewer's comments. We have reorganized Introduction and methods section as suggested by the reviewer. We have split 3.2 into sections 3.2 and 3.3 as recommended by the reviewer.

(1) I suggest defining dry deposition early on in the abstract and introduction. Which processes are considered dry deposition? Are they all dependent on concentration to first order (L24 states how it is calculated, could you state what it represents)? Why is it 'dry' vs 'wet', and how relevant is the distinction for different gases/processes? For many gases with uptake fluxes into the biosphere, the term 'deposition' is a bit misleading, because rather than depositing like a dust particle or aerosol, gases are often taken up by gradient-driven biochemical reactions that vary in time and space in ways that are not consistent with the idea of simple deposition on a surface. I understand the historical use of this term, and my suggestions here is just to add more description of the involved processes that are referred to collectively as dry deposition earlier in the manuscript.

In the revised manuscript, dry deposition is defined in the abstract and introduction. Major factors affecting dry deposition process is briefly mentioned (which includes meteorological, biological and chemical factors). The concept of dry versus wet deposition has been added in the introduction. Here are some revised texts:

In Abstract: "Dry deposition process refers to flux loss of an atmospheric pollutant due to uptake of the pollutant by the earth's surfaces including vegetation and underlying soil and any other surface types."

"$V_d$, the latter is a variable that needs to be highly empirically parameterized due to too many meteorological, biological and chemical factors affecting this process."

In Introduction: "In mass continuity equation of a chemistry transport model (CTM), atmospheric deposition is calculated separately for dry and wet deposition fluxes. Dry deposition refers to the removal process through which pollutants are taken up by the earth's surface, and this process, while being quite slow, is a continuous process happening all the time, even during precipitation. In contrast, wet deposition is fast but episodic, and pollutants need to be first incorporated into hydrometeors before being delivered to the surface via precipitation."

abstract: Instead of relying on the citation of Zhang et al. (2003) to describe the nature and extent of the dry deposition scheme, please be more descriptive in this second sentence of the abstract and describe in simple terms what dry deposition processes are, what the Zhang version includes, and was anything in the scheme fundamentally changed except adding new gas species?

We have modified the first half of the abstract to include descriptions of the dry deposition process and how the model of Zhang et al. (2003) is modified in this study to include additional oVOCs. The revised text reads: "Dry deposition process refers to flux loss of an atmospheric pollutant due to uptake of the pollutant by the earth's surfaces including vegetation and underlying soil and any other surface types. In chemistry transport models (CTMs), dry deposition flux of a chemical species is typically calculated as the product of its surface-layer concentration and its dry deposition velocity ($V_d$), the latter is a variable that needs to be highly empirically parameterized due to too many meteorological, biological and chemical factors affecting this process. The gaseous dry deposition scheme of Zhang et al. (2003) parameterize $V_d$ for 31 inorganic and organic gaseous species. The present study extends the scheme of Zhang et al. (2003) to include additional 12 oxidized volatile organic compounds (oVOCs) and hydrogen cyanide (HCN), while keeping the original model structure and formulas, to meet the demand of CTMs with increasing complexity. Model parameters for these additional chemical species are empirically chosen based on their physicochemical properties, namely the effective Henry's law constants and oxidizing capacities."

Methods: Is dry deposition to canopy/vegetation only or does it also include soil? This information should be given in introduction, instead of only being fist mentioned in L119.

The following explanation has been added in the first paragraph of Introduction: "In most $V_d$ formulations, turbulent and diffusion effects are parameterized as aerodynamic and quasi-laminar resistance, respectively, above and sometimes also inside the canopy. Uptake effects by canopies and underlying soils and any other surface types are parameterized as canopy (or surface) resistance, which include several flux pathways such as to stomatal, cuticle and soil."

L90 give the model equations/formulation in this paper earlier in the methods, instead of relying on the reader accessing Zhang et al. 2002 or waiting to L134. Does H* enter into the model formulation directly, or just inform the parameterization of alpha and beta? The 'scaling parameter' terminology is helpful for understanding these factors in Table 1, and could be used in the text to make their meaning clearer. Give a formula for how you scaled alpha for oVOCs relative to that of SO2.

We have reorganized section 2 into the following three sub-sections: *2.1 Brief description of the $V_d$ formulation; 2.2 Extension of the scheme to additional chemical species; and 2.3. Field flux*

*data.* Materials are adjusted accordingly. After this reorganization, formulas appeared first as recommended by the reviewer.

*H\* is not used directly in any formula, instead, it is used for choosing the alpha value. This has been describe clearly in section 2.2: "Initial α values were first given based on the relative magnitudes of H\* of all the chemical species and that of $SO_2$" "When adjusting α and β values, two rules were first applied: (1) the trends in α (or β) values between different chemical species should be consistent with the trends of their log(H\*) (or oxidizing capacity) (see Figure S1 for the finalized α versus log(H\*)); and (2) modeled mean and median nighttime $V_d$ should be mostly within a factor of 2.0 of the measured values."*

L196-201: this is really helpful background information for the model that might be more useful in the introduction or methods section, rather than only being presented in the results. Same comment regarding the introduction material on stomatal conductance and transpiration fluxes.

After careful considerations we feel that it is not a good space in the Introduction section to discuss model theories. This is because the focus of the present study is to extend the model to additional chemical species without modifying the model structure or theory. Thus, the introduction section discusses the basic concept of dry deposition, the current knowledge status of oVOCs dry deposition, and the approach of extending the model to include additional oVOCs. Discussing too much details of model theory (such as including stomatal update process) in the introduction will loose the major focus of the study. We do have added a simple description of the model theory that mentions stomatal uptake in Introduction, which reads: "Uptake effects by canopies and underlying soils and any other surface types are parameterized as canopy (or surface) resistance, which include several flux pathways such as to stomatal, cuticle and soil."

Because of the same reason (no change in model structure or theory), we feel there is no need to add detailed description of the model theory in the Methods section either, especially considering such theory is well known in the air-surface exchange scientific community. The theory was only briefly mentioned in the Results section where it is needed to explain the results (as an introductory sentence in each topical discussion).

L230: give some examples of what other processes can affect deposition earlier in this paragraph, rather than leaving it to the end as "leaf cuticle and ground (more specifically soil/litter) or reactions within and near canopy".

As mentioned in several responses above, we have added brief statements in Abstract and Introduction, mentioning the many process affecting deposition. For this particularly comment, we have further added this statement: "All of these flux pathways can be simultaneously affected by meteorological, biological and chemical factors, most of which cannot be explicitly considered and thus are highly empirically parameterized in dry deposition models."

(2) Appropriate context for the current state of understanding is lacking. Give examples of the oVOCs relevant to this paper and their properties before L31, so we can understand how suitable it may be to use SO2 or O3 as references. For example, you state that the O3 reaction with oVOCs should depend on chemical structure—please describe this in more detail and list the oVOCs you will consider beforehand so we have context. L50-53 on HCN feels like an orphan sentence—suggest to make a different paragraph where oVOC and HCN properties are discussed together. Define IEPOX in Table 1 or text referring to it the first time.

Field flux measurement data on oVOC are extremely rare, and currently we do not have a very good understanding on the deposition process of oVOCs. The approaching of using $SO_2$ and $O_3$ as base species for scaling the non-stomatal uptake of other chemical species including oVOCs has been used in several widely used community dry deposition schemes (e.g., Wesley et al, 1989; Zhang et al., 2003). The impact of reactivity on dry deposition cannot be quantitatively assessed or applied in dry deposition schemes due to the limited knowledge at present. It is their relative reactivity that are important for choosing model parameters (such as beta used in this model). All related chemical reactions are listed in Table S2 of Supporting Information. Because of these reasons, we do not have much to add to the existing discussion in Introduction regarding their reactivity.

I would suggest re-writing L32-L50 to make more general statements that are illustrated by the discrepancy between Zhang and Karl studies, rather than being so specific about these papers. Otherwise, the introduction reads more like a discussion and feels very narrow, and the expectation (L43) is written more like a conjecture. Give a range of ratios for oVOCs and O3 Vd so we can more clearly compare how they differ.

Here we tried to convey these facts: There are only two studies reporting higher $V_d$ for oVOC than those predicted by existing dry deposition schemes. Discussing some details revealed in one of the studies (Karl et al., 2010) can then illustrate (i) the possibly of chemical effect on dry deposition flux, and (ii) the potential large uncertainty in their flux data (which was not directly measured, but generated from concentration gradient using a model). We have modified the introductory sentences so that theses discussions can be read in a logic way. The revised text reads: "In these existing schemes, $V_d$ of most oVOCs were on the similar magnitude to or slightly smaller than that of $V_d$ of $O_3$. However, higher daytime $V_d$ values for certain oVOCs than predicted by existing schemes were reported lately by two studies (Karl et al. 2010; Nguyen et al., 2015). In one study Karl et al. (2010)……"

(3) L64: what specifically do you mean by 'this approach'? The last sentence ends with the inability of the model to match high daytime values… making it unclear what approach you are referring to because the last description of the approach was a negative one. Give model context. L65, consideration 1: the introduction did not give sufficient context for what you mean here, please describe this more clearly. It is unclear how your approach addresses these considerations.

We have modified and moved this part to the third paragraph of section *2.2 Extension of the scheme to additional chemical species*. This particularly sentence has been changed to this: "Model parameters chosen for the additional oVOCs and HCN can produce the magnitude of nighttime $V_d$ for nearly all the chemical species, but inevitably underpredicted daytime $V_d$ for several oVOCs species with very high measured daytime $V_d$ values."

(4) End of introduction. L71-74: very general, long sentence. Reads more like a conclusion/future outlook. As does most of this paragraph. Much of this final paragraph does not appear to be relevant to the specific approach taken in this study, so it belongs earlier in the introduction or maybe in the conclusions. Be specific about what the contribution of your effort here is.

We have moved some of the materials in this paragraph to the place before describing our own approach in the Introduction, and rewritten the last two paragraphs of Introduction accordingly so the material can be presented in a logic way. We also feel that it may fit into conclusion, but we need such a discussion to support why we choose our approach, so we think it is better to

keep it in Introduction, and in a place before describing our approach. The revised text reads: "To meet the demands of modeling a large number of organic compounds in CTMs (Kelly et al., 2019; Moussa et al., 2016; Paulot et al., 2018; Pye et al., 2015; Xie et al., 2013), existing or newly developed air-surface exchange/dry deposition schemes need to be expanded to include additional oVOCs. At this stage with very limited knowledge on oVOC $V_d$, air-surface exchange models based on various theoretical and/or measurement approaches should be developed, so that these models can be made available to the scientific community where such models are urgently needed, and for future evaluation and improvement should more flux measurements become available. For example, Nguyen et al. (2015) modified the Wesely (1989) scheme to fit the flux data. A more sophisticated model, with a bottom-up approach, was adopted in Nizzetto and Perlinger (2012) to handle air-canopy exchange of semivolatile organic compounds.

The original dry deposition scheme of Zhang et al. (2003) includes 9 inorganic species and 22 organic species. Most of these 22 organic species are oVOCs formed from oxidation of nonmethane hydrocarbons. To take advantage of the recent flux dataset of a large number of oVOCs and HCN collected over a temperate forest (Nguyen et al., 2015), the present study extends the Zhang et al. (2003) scheme by including 12 additional oVOC species and HCN while keeping the same original model structure and theory. These additional oVOCs include hydroxymethyl hydroperoxide, peroxyacetic acid, organic hydroxy nitrates, and other multifunctional species that are mainly formed from the oxidation of biogenic VOCs (e.g., isoprene and monoterpenes). Model parameters for these newly-included species are theoretically constrained based on the effective Henry's law constants and oxidizing capacities of the individual species and by considering the measured $V_d$ values as well. Such an approach provides a top-down determination of $V_d$ through comparison with measured (bottom-up) fluxes. Model-measurement comparison is conducted for $V_d$ as well as resistance components/uptake pathways, results from which identify the major causes of model-measurement discrepancies. This study provides a computer code that is potentially useful for CTMs handling these oVOCs."

(5) Since the measurement of VOCs is highly dependence on the instruments used, state the instrumentation used to measure the 13 VOCs, HCN, H2O2, and HNO3 in this paper instead of relying on the Nguyen et al., 2015 paper alone.

We have added a sentence to specify the instrument of oVOCs measurement, which reads "Mixing ratios of gas-phase compounds were measured with negative-ion chemical ionization mass spectrometry (CIMS) at 8 Hz or faster."

(6) Were there ever net emissions of these compounds from the ecosystem, and how did that factor into Vd calculations? Please comment on what role soil uptake might or might not play in the large observed residual uptake of oVOCs during daytime dry conditions.

As mentioned above, field flux measurement data on oVOC are extremely rare. For most oVOCs considered here, dry deposition should dominate over emission so only net dry deposition is considered in the model.

Soil uptake should not be a dominate non-stomatal or residual uptake of oVOCs as it can be limited by the weak in-canopy turbulence especially for a closed canopy such as the forest site in this study.

(7) How is Gns calculated, and how does that differ from Gresidual? An equation is only given for the latter (L231). The difference between the two, and why we must assume that Gns terms are correctly estimated is not clear. Please elaborate and justify.

*$G_{ns}$ is calculated according to Eq (3) in the manuscript ($G_{ns} = 1/R_{ns} = 1/(R_{ac}+R_g) + 1/R_{cut}$). The $R_{ac}$, $R_g$, and $R_{cut}$ terms are from the modeling result of the Zhang scheme. As described in the manuscript, $G_{residual}$ is estimated as $[V_d^{-1} – (R_a + R_b)]^{-1} – (R_s+ R_m)^{-1}$ where $V_d$ is from the eddy-covariance measurements, $R_s$ is calculated by the Penman-Monteith equation using measured water vapor flux, $R_a$ and $R_b$ rely on conventional micrometeorological approaches driven by measured meteorology (e.g., $u_*$). These formulas are clearly presented in the manuscript.*

**Minor comments:**

L33: cite the existing schemes or describe how they are different from what you do here.

*The existing schemes refer to those mentioned in the preceding sentence (Wesely, 1989; Zhang et al., 2003). We have added a sentence before this sentence and we think it is now clear. The revised text reads: "Due to the lack of field flux data of oVOCs, $V_d$ of these species is typically parameterized based on physicochemical properties, taking $SO_2$ and $O_3$ as references (Wesely, 1989; Zhang et al., 2003). In these existing schemes, $V_d$ of most oVOCs were on the similar magnitude to or slightly smaller than that of $V_d$ of $O_3$. However, higher daytime $V_d$ values for certain oVOCs than predicted by these schemes were reported lately by two studies (Karl et al. 2010; Nguyen et al., 2015)"*

L54: be more specific, what does 'community demands' mean?

*We have modified the sentence to this: "To meet the demands of modeling a large number of organic compounds in CTMs (Kelly et al., 2019; Moussa et al., 2016; Paulot et al., 2018; Pye et al., 2015; Xie et al., 2013), existing or newly developed air-surface exchange/dry deposition schemes need to be expanded to include additonal oVOCs."*

L263: had instead of have

*Corrected.*

L270: discussion starts?

*This paragraph starts interpolating the model evaluation results and discusses the possible causes of the model-measurement discrepancies using knowledge from literature. We have split this section into two separate sections for easy reading. The additional section is given a title of "3.3. Fast chemical reactions as potential causes of the daytime model-measurement discrepancies."*

L110: specify if you mean formic acid is the only species available in the original Model.

*The original Zhang scheme (Zhang et al., 2003) includes 9 inorganic species and 22 organic species. Most of these 22 organic species are also oVOCs produced from nonmethane hydrocarbons (NMHCs) oxidation process. In this study, we extended the model to include 12 new oVOCs for which flux measurements over a temperate forest were available. These additional oVOCs include hydroxymethyl hydroperoxide, peroxyacetic acid, organic hydroxy nitrates, and other multifunctional species and are mainly formed from the oxidation of biogenic VOCs (e.g., isoprene and monoterpenes). Formic acid is the only overlapped species between the original model and the measurement data set. We have added the above information in the Introduction.*

---

## Author Comment (AC5)

**Response to Reviewer #2**

We greatly appreciate the reviewer for providing valuable comments on our manuscript, which have helped us improve the paper quality. We have addressed all of the comments carefully as detailed below. The original comments are in black and our replies are in blue.

This is a generally well-written paper about a difficult scientific topic. The authors document how a well-know dry-deposition model can be extended to treat additional oVOC species. The authors are honest about limitations, and have good explanations for most of the issues. I do have concerns about the assumptions concerning Gns versus Gresidual, as well as some other points as given below. As long as these can be addressed satisfactory then the article, and in particular the changes to the deposition code, will be a useful addition to the literature.

**General**
The assumption that Gns is "correctly estimated" (L236) when looking at the Gresidual is of course a major problem. As noted by for example Massman (2004), or Cape et al (2009), these non-stomatal terms are very uncertain even for ozone. I would like to see a more thorough assessment of this issue.

"correctly estimated" should be replaced with "estimated with reasonable accuracy". We agree with the reviewer that the existing formulas for estimating non-stomatal terms have very large uncertainties. Compared to the other existing dry deposition schemes, the one used in Zhang et al. (2003) is actually the only one considering several key meteorological factors. For example, in Wesely (1989), constant values were used for this term for a specific land use. The uncertainties in individual resistance terms have been thoroughly discussed in Wu et al. (1028), which support this assumption: $G_{residual}$ estimated using the formula $[V_d^{-1} – (R_a + R_b)]^{-1} – (R_s + R_m)^{-1}$ is meaningful." We have modified this part to this: "The uncertainties in individual resistance terms of Zhang et al. (2003) and several other dry deposition schemes have been thoroughly assessed by Wu et al. (2018), from which we believe $G_{residual}$ estimated using the above formula is meaningful although with large uncertainties. The estimated $G_{residual}$ can provide…"

Also in this respect, the model assumes that surfaces are either wet or dry. Of course, the real world shows a high degree of variability, and it can be difficult to predict the thickness or coverage of moisture films on leaves (e.g. Wichink Kruit et al., 2008). How can the authors be confident that their Gns is correct when such basic factors as leaf-wetness (and its impacts on aqueous/surface reactions) are so hard to deal with?

I would have liked to see some analysis of the results with RH (or deficit D) as the driving variable, rather than just wet/dry.

In the figure below, we analyzed the nighttime $G_{residual}$ and $G_{ns}$ under different RH conditions (similar to Figure 3 in the manuscript). Both $G_{residual}$ and $G_{ns}$ tended to increase with higher RH, which is consistent with our findings with dry/wet surface at night.

[Figure]

Figure. Observation-based residual conductance ($G_{residual}$) and the modeled nonstomatal conductance ($G_{ns}$) under different humidity conditions during nighttime. The sample sizes for RH <75, 75-90, and >90 are 20, 50, and 58, respectively. The box covers the 25-75[th] percentiles range with median (horizontal line) and the arithmetical mean (filled dot) of the 25-75[th] percentiles data also shown inside the box.

I would also have liked to see some indication and better discussion of the uncertainty of the flux measurements. These uncertainties are substantial, and presumably contribute to some of the differences seen in e.g. Fig. 4.

Nguyen et al. (2015) provided some discussions on the measurement uncertainties. For example, the Table S1 of Nguyen et al. (2015) showed that the sensor sensitivity uncertainties ranged from 20-50% for the oVOC species. We agree that the measurement uncertainties could contribute to the model-measurement discrepancies showed in this study, but the data we have are not enough for assessing the uncertainties in a quantitatively way.

When modeling the deposition of organic compounds, I wonder why water is the only solvent being considered when calculating Rns? Much of the SOA modeling conducted with CTMs assumes indeed that SOA species are absorbed in the organic rather than the water component of

the particle. Perhaps complex thermodynamic models (e.g. Zuend et al, 2011) are required to cope with the deposition (or bi-directional exchange) of these compounds?

The organic matters could be an effective solvent for the oVOC compounds. Some studies in literature (e.g., Nizzetto and Perlinger, 2012; Wu et al., 2003) used the octanol-air partitioning coefficients to parameterize the absorption of the organic compounds in organic solvent. Currently the Zhang scheme doesn't include the octanol-air partitioning coefficients for the deposition compounds. In the future, new scheme can be further developed by including the octanol-air partitioning coefficients and coupling with complex thermodynamic models once the proper parameterizations and reliable parameter values are available. As we recommended in the Introduction: "At this stage with very limited knowledge on oVOC $V_d$, air-surface exchange models based on various theoretical and/or measurement approaches should be developed, so that these models can be made available to the scientific community where such models are urgently needed, and for future evaluation and improvement should more flux measurements become available."

Terminology: I must admit I don't like anybody referring to their own code as "the Model", with capital M, which makes it sound like it is the ultimate reference. Better to say "the model" or "the deposition model" or something similar.

The term "the Model" has been removed throughout the manuscript.

**Other comments**

L50: The sentence about HCN doesn't seem to fit with the rest of this paragraph, or the oVOC theme in general. Start a new paragraph maybe?

A separate paragraph is used for HCN discussion in the revised manuscript.

L117-, Do equations 2-3 ascribed to Wu et al. 2018 differ from those of equation 4 which is ascribed to Zhang et al 2002? (It is a little confusing here what is meant by "the Model", when the latter was stated on L108 to be Zhang et al 2003!)

Zhang et al. (2003) is an updated version of Zhang et al. (2002), where the non-stomatal resistance parameterizations were updated while the stomatal resistance sub-module was kept the same. $R_a$ and $R_b$ formulas were not provided in either Zhang et al. (2002) or Zhang et al. (2003) because various but very similar formulas are available in literature. In summary, the details of the $R_s$ formulas were described in Zhang et al. (2002), $R_{ns}$ formulas in Zhang et al. (2003), and $R_a$ and $R_b$ formulas in Wu et al. (2018). We thus have to cite different references for these resistance formulas. In the revised manuscript, we have removed the citation of Zhang et al. (2002) and Wu et al. (2018) in two places to avoid confusion, and instead, we have added this statement at the end of section 2.1 for clarification: "Details of the $R_s$ related formulas were described in Zhang et al. (2002), $R_{ns}$ related formulas in Zhang et al. (2003), and $R_a$ and $R_b$ formulas in Wu et al. (2018)."

L179-, Fig.1. The authors discuss the discrepancy in HNO3 Vd for hours 19-23. but not why Vd in hours 0-3 is so very different. What happens at midnight that could change Vd?

The figure below presents the averaged diel variation of measured friction velocity ($u_*$) which showed lower $u_*$ at the early night (19-23) than the late night (0-3), consistent with the trend of the measured $V_d(HNO_3)$. One possible reason for the large model-measurement discrepancies in $V_d$ for $HNO_3$ could be the poor performance of the $R_a$ parametrization under low $u_*$ conditions.

[Figure]

Figure. Averaged diel variations of measured friction velocity ($u_*$) at the CTR site during the study period.

L196- I agree with ref #1 that this material is background and should come earlier.

This paragraph describes the method for calculating the observation-based stomatal conductance so we can compare the observation-based and modeled stomatal conductance. Materials here are closely linked with the model-measurement comparison discussion presented in this section. We thus prefer not to move it to the Introduction.

As we responded to reviewer #1 on a similar comment: "The focus of the present study is to extend the model to additional chemical species without modifying the model structure or theory. Thus, the introduction section discusses the basic concept of dry deposition, the current knowledge status of oVOCs dry deposition, and the approach of extending the model to include additional oVOCs. Discussing too much details of model theory (such as including stomatal uptake process) in the introduction will loose the major focus of the study."

L214. Please add a ref to Fig. 2 here, so the reader knows what you are talking about.

We have added this: "As shown in Figure 2," at the beginning of the paragraph.

L216 claims that "the Jarvis" model is used, but are the Gs equations and parameters as used here (in "the Model") identical to those used in the 1976 Jarvis paper? If not, rephrase

We have rephrased it to "the Jarvis-type".

L223. Again, is the stress function used here identical to that from Jarvis 1976? In any case, all such stress functions are very sensitive to the very uncertain methods used to estimate soil water potential (or other metrics, e.g. Buker et al, 2012)

No, the stress functions mostly follow the SiB1 model (Sellers and Dorman, 1987) and the details can be found in Brook et al. (1999). In the case of this study, the stress factor from water vapor pressure deficit (VPD) was much lower than the other stress factors around noon and thus dominated the reduction of noon-time canopy stomatal conductance. Here we have also rephrased it to "the Jarvis-type".

L241-242. The authors say that during night-time the "canopy surface was dry (no dew)", but presumably RH was high and some surface moisture was possible.

We agree that high RH at night could result in microscale water films on the canopy surfaces (invisible wetness). The Zhang et al. (2003) scheme follows the approach of Janssen and Romer (1991) to predict the occurrence of dew, which depends on wind speed, temperature and dew point temperature and this has been described in Brook et al. (1999). The prediction of microscale water films is much more uncertain and currently the Zhang's scheme does not include such a parameterization. In a practical way, we classified the surface without predicted dew as dry condition and the surfaces with dew as wet condition. As shown in Figure 3, the nonstomatal conductance exhibited significant differences between the dry and wet conditions. The influence of the microscale wetness due to high RH is expected to be minimal and will not change any conclusions in this study.

L289. The paper states that the measured flux at night-time should better represent non-stomatal surface uptake, but it is also true that fluxes are very hard to measure at night-time. A brief discussion of this, and its implications, is warranted in the paper. (There are some comments starting on L330 that help in some regard, but these suggest that essentially one cannot trust the night-time Vd calculations; hence no relation with Gns can be established?)

We are aware of that the uncertainties in the measured fluxes are even larger in nighttime than daytime. This is the case even for the most commonly studies species such as $O_3$, $SO_2$, and some nitrogen species with rich flux data set, as also noted above by this reviewer in his/her general comment. That is why we provided a brief discussion/recommendation in L330 in the original manuscript. These large uncertainties making it difficult to obtain a good correlation between the modeled $G_{ns}$ and measured nighttime flux. Nevertheless, we believe the magnitude of the campaign-averaged measured nighttime flux should be reasonable, so we aim to model $G_{ns}$ to be within a factor of 2 of the measured flux on campaign-average time scale. Since this is a common issue to nearly all the chemical species (not just applying to oVOCs studied here), we feel we do not have any extra information to add, other than what has already been presented in L330 and below.

L303. So, what do the chemists tells about the reactivity of PAA versus HAC? I suggest giving some reaction rates and time-scales with OH, O3 and NO3.

According to Wesley (1989), oxidizing capacities can be described by redox reactions. We have generated related parameters and the details are provided in Table S2 of the Supporting Information. Based on their $pe^0(W)$ values, PAA is indeed more reactive than HAC (0.16 versus -2.35 $pe^0(W)$). We have added this statement in the revised manuscript where PAA and HAC are

compared: "The reactivity parameters listed in Table S2 in Supporting Information also suggest PAA is more reactive than HAC."

L395. Should give the doi

We have modified this statement to this: "The computer code and data used in this study can be obtained from contacting the corresponding author. The code is also available from (DOI:10.5281/zenodo.4697426): https://zenodo.org/record/4697426#.YHmzu5-Sk2w"

**References mentioned in this response:**

Brook, J., Zhang, L., Digiovanni, F., Padro, J., 1999. Description and evaluation of a model for routine estimates of air pollutant dry deposition over North America. Part I: Model devlopment. Atmospheric Environment 33, 5037–5051.

Janssen, L.H.J.M., Romer, F.G., 1991. The frequency and duration of dew occurrence over a year. Tellus 43B, 408-419.

Nguyen, T. B., Crounse, J. D., Teng, A. P., Clair, J. M. S., Paulot, F., Wolfe, G. M., et al. (2015). Rapid deposition of oxidized biogenic compounds to a temperate forest. Proceedings of the National Academy of Sciences, 112(5), E392-E401.

Nizzetto, L. & Perlinger, J.A. (2012). Climatic, biological, and land cover controls on the exchange of gas phase semivolatile chemical pollutants between forest canopies and the atmosphere. Environmental Science & Technology, 46(5), 2699-2707.

Sellers, P.J., Dorman, J.L., 1987. Testing the simple biosphere model (SiB) using point micrometeorological and biophysical data. Journal Climate Applied Metrology 26, 622-651.

Wesely, M. (1989). Parameterization of surface resistances to gaseous dry deposition in regional-scale numerical models. *Atmospheric Environment, 23*(6), 1293-1304.

Wu, Y., B. Brashers, P. Finkelstein, and J. Pleim, A multilayer biochemical dry deposition model, 1 Model formulation, J. Geophys., 107, doi:10.1029/2002JD002293, in press, 2002.

Wu, Z. Y., Schwede, D. B., Vet, R., Walker, J. T., Shaw, M., Staebler, R., et al. (2018). Evaluation and intercomparison of five North American dry deposition algorithms at a mixed forest site. Journal of Advances in Modeling Earth Systems, 10(7), 1571-1586.

Zhang, L., Brook, J., & Vet, R. (2003). A revised parameterization for gaseous dry deposition in air-quality models. Atmospheric Chemistry and Physics, 3(6), 2067-2082.

Zhang, L., Moran, M. D., Makar, P. A., Brook, J. R., & Gong, S. (2002). Modelling gaseous dry deposition in AURAMS: a unified regional air-quality modelling system. Atmospheric Environment, 36(3), 537-560.